# Detecting chromatin state alterations in PBMCs associated with Type 2 Diabetes Mellitus

Maryam Moazeni Afarani[1,8], Rajshikhar Gupta[1,2,7,8], Caroline Uhler [3,4], Issa Fetian[5,6] & GV Shivashankar[1,2] ✉

## Abstract

**Background** Type 2 Diabetes Mellitus (T2DM) involves patient-specific immune dysfunction not addressed by current diabetes management methods. Traditional methods to assess functional changes to PBMCs are too costly and time-intensive for routine use.
**Methods** We studied 57 individuals across healthy, prediabetic, and diabetic stages, performing chromatin imaging of live peripheral blood mononuclear cells (PBMCs) using a microfluidic imaging assay.
**Results** Here, we present an orthogonal and adjunct chromatin imaging-based assay to monitor alterations to PBMCs during T2DM progression. By applying representation learning on chromatin images, we identified distinct PBMC clusters, with specific subpopulations enriched at different T2DM stages. We found that the levels of certain PBMC subsets are predictive of T2DM. Additionally, we observed significant changes in nuclear and chromatin mechanics in diabetic individuals compared to prediabetic and healthy individuals, along with decreased Lamin A/C expression and increased cellular activation in diabetic PBMCs.
**Conclusions** Collectively, this study demonstrated a cost-effective and scalable solution for the routine monitoring of T2DM using PBMC chromatin biomarkers.

## Plain language summary

In this study, we examined blood samples from 57 people, including healthy individuals, people with prediabetes, and people with type 2 diabetes (T2DM). Using a microscopy-based method to look at how DNA is organized inside immune cells, we found clear differences between these groups. We observed that certain immune cell types show distinct changes as diabetes progresses. These changes were specific to individual patients and were more pronounced in people with T2DM compared to those with prediabetes or no diabetes. In people with T2DM, immune cells also showed signs of increased activation, reduced levels of a structural protein called Lamin, and altered mechanical properties, suggesting that the cells are under stress and behaving differently. Overall, our results show that chromatin imaging can capture meaningful changes in immune cells linked to diabetes progression. Because this approach is relatively affordable and scalable, it has the potential to be used in clinical settings to help monitor disease progression and support more personalized management of type 2 diabetes.

While recent advances in diabetes care and management technology have enhanced miniaturization and accessibility, such as insulin delivery systems and blood glucose monitors[1,2], they do not address patient-to-patient variability in Type 2 Diabetes Mellitus (T2DM) related immune dysfunction. In T2DM, the inflammatory environment-characterized by hyperglycemia,

elevated free fatty acids, and cytokines-pushes immune cells toward glycolytic, pro-inflammatory phenotypes, resulting in subpopulation imbalance[3–7]. For instance, T2DM onset is associated with the accumulation of pro-inflammatory T cell subsets, such as Th1 and Th17 cells, and a reduction in anti-inflammatory regulatory T cells in the adipose tissue due to altered

[1]Laboratory of Multiscale Bioimaging, Paul Scherrer Institut, Villigen, Aargau, Switzerland. [2]Department of Health Sciences and Technology, ETH Zürich, Zürich, Switzerland. [3]Laboratory of Information and Decision Systems, MIT, Cambridge, MA, USA. [4]Eric and Wendy Schmidt Center, Broad Institute of MIT and Harvard, Cambridge, MA, USA. [5]Hausarztpraxis MZ Brugg, Brugg, Aargau, Switzerland. [6]Universität Basel, Basel, Switzerland. [7]Present address: Eric and Wendy Schmidt Center, Broad Institute of MIT and Harvard, Cambridge, MA, USA. [8]These authors contributed equally: Maryam Moazeni Afarani, Rajshikhar Gupta. ✉e-mail: gshivasha@ethz.ch

insulin signaling[8]. Changes in circulating and resident immune cell subpopulations in the adipose tissue during the onset of T2DM drive compositional remodeling of the visceral adipose tissue (VAT) microenvironment, skewing resident macrophages toward a pro-inflammatory M1 phenotype, which secrete cytokines such as TNF-$\alpha$, IL-6, and IL-1$\beta$[9]. Together with infiltrating subpopulations of NK and B lymphocytes[10,11], these M1 macrophages amplify inflammation in the microenvironment of VAT and other organs, leading to patient-specific comorbidities[12,13]. In the liver, adipose-derived cytokines and immune cell trafficking activate Kupffer cells and recruit monocytes and T cells, promoting NASH and fibrosis in specific patient cohorts[14]. Similarly, in certain patient subpopulations, elevated levels of VAT-derived mediators (e.g., MCP-1, TNF-$\alpha$, IL-6, IL-18) drive macrophage and T-cell infiltration into glomeruli and tubules, fueling chronic inflammation and fibrosis that characterize diabetic nephropathy[15]. Inclusively, in pancreatic islets, macrophage-derived IL-1$\beta$ and TNF-$\alpha$ impair insulin secretion and reduce $\beta$-cell survival, linking adipose inflammation to endocrine dysfunction in T2DM[16]. Beyond shifts in immune cell subpopulation, the enhanced glycolytic activity, inflammatory cytokines, and elevated ROS production observed in T2DM drive premature T cell senescence and activation[5,17,18]. These functionally altered immune cells fail to effectively extravasate into target tissues[17,19,20] due to cell-intrinsic metabolic defects contributing to surveillance deficits and secrete pro-inflammatory cytokines (like IFN-$\gamma$ and TNF-$\alpha$) contributing to the disease progression[21,22]. Collectively, these findings suggest that altered and dysfunctional immune cell recirculation and local inflammatory cues from VAT integrate signals between metabolic organs and perpetuate disease progression. However, in this context, major gaps exist in detecting immune cell subpopulation stratification during the onset of T2DM in the clinically relevant setting.

Single-cell multi-omic studies of peripheral blood mononuclear cells (PBMCs) from T2DM patients versus healthy controls have unveiled shifts in immune cell subsets and activation states[23–26]. The transcriptional response and functional states of immune cells change during T2DM progression, which are strongly influenced by factors such as patient age[27], existing co-morbidities[28,29], therapeutic interventions (e.g., drug treatments)[30], and diet[31]. While tracking immune cell subpopulation stratification can be used to monitor T2DM, this requires methods such as single-cell sequencing[32] and proteomics[33], which remain too costly, labor-intensive, and impractical for a clinical setting. We recently demonstrated that chromatin imaging of liquid biopsies can provide novel ways of detecting and evaluating interventional efficacy in various cancer types[34,35]; These findings have motivated us to explore whether chromatin imaging-based assays of immune cells could overcome the limitations of existing techniques in the context of metabolic disorders.

In this cross-sectional study, we select a cohort of 57 individuals spanning healthy, prediabetic, and diabetic stages. Using chromatin imaging, we identify cell-specific enrichment patterns associated with distinct stages of T2DM progression. Our findings indicate that specific PBMC subpopulations reveal patient-specific alterations in immune cells in individuals with T2DM compared to prediabetic and healthy controls. In addition, we observe significant changes in the mechanical properties, a decrease in Lamin expression, and an increase in cellular activation in the PBMCs from T2DM patients. In summary, we demonstrate that our chromatin imaging assay can be an affordable and scalable approach for potential clinical use in monitoring PBMC chromatin alterations relevant to T2DM management and personalized disease tracking.

## Methods
### Experimental design
**Patient sample collection.** The study protocol was approved by the Ethics Committee of Switzerland (approval number EKNZ 2021-02229), the medical center Brugg, Switzerland, and the Paul Scherrer Institute. The protocol adheres to all relevant ethical regulations, including those specified in the Declaration of Helsinki. According to the detailed study protocol, available under the approval number mentioned above, all samples were collected with informed written consent from the individuals who participated in this study. Blood samples were collected from individuals who were healthy, prediabetic,

and diabetic. The initial diagnosis was performed using the combination of gold standard clinical methods such as fasting Plasma Glucose (FPG), oral glucose tolerance test (OGTT), and glycated hemoglobin (HbA1c). We have shown the primary, secondary, and tertiary diagnoses for each individual in the Supplementary Table S1. The healthy donors had no significant medical history of chronic disorders. A total of 12 ml of blood was collected in Ethylenediaminetetraacetic acid (EDTA)-coated tubes from all the individuals who participated in this study. Blood samples were stored in a refrigerator at 4 °C immediately after collection. The samples were then placed in double-layer protection, with an absorption pad inside an additional tube to prevent any leakage during transport. Finally, they were placed in a cold box and transferred from the medical center of Brugg to the Paul Scherrer Institute. All blood samples were processed within 24 h of collection.

**PBMCs isolation.** All blood cell isolation procedures were conducted under biosafety level 2 (BSL-2) conditions per the institute regulations, and isolation was performed within a laminar flow cabinet to prevent aerosol generation. PBMCs were isolated using a standard density gradient centrifugation protocol. Briefly, blood samples were diluted 2:1 with phosphate-buffered saline (PBS) containing 2% fetal bovine serum (FBS) and added to Lymphoprep density gradient medium (STEMCELL Technologies, catalog no. 07811) in SepMate¨−50 tubes. The tubes were centrifuged at 1190 rpm for 20 min at room temperature, allowing the PBMCs to accumulate in a layer between plasma and density gradient media. The plasma and PBMC layers were transferred to a 50 ml Falcon tube and centrifuged again at 3000 rpm for 10 min. The PBMCs were then transferred to new tubes and washed twice with PBS containing 2% FBS. Approximately $6 \times 10^5$ cells were re-suspended and cryo-preserved in cryo-preservation media containing RPMI (Roswell Park Memorial Institute) 1640 medium (Thermo Fisher), 20% FBS, and 10% dimethyl sulfoxide (DMSO, Sigma Aldrich) and stored at −80 °C for microfluidic experiments. Around 10% of the whole isolated PBMCs ($6 \times 10^5$ cells) were fixed with 4% paraformaldehyde (PFA, Sigma Aldrich) for 20 min and stored at −20 °C for subsequent immunostaining and microscopy. The 90 percent remaining PBMCs were kept alive and stored at −80 °C as a backup for short-term experimental use.

**Photolithography.** The microfluidic pattern was created with Klayout 0.27.4 software and subsequently produced using the DWL 66+ system from Heidelberg Instruments Mikrotechnik GmbH (Heidelberg, Germany), operating at a 405 nm (h-line) wavelength. The write mode 3 (WM III) was used, with laser writing parameters set at optical transmission 20%, focus 0%, intensity 100%, filter 50%, and laser power 64 mW (dose on substrate 135 mJ.cm$^{-2}$). A Laser Scanning Confocal Microscope (LSCM, VK-X3100, Keyence Corporation, Japan) was utilized as a non-destructive method to measure the height of the resist structure, which lacked any antistatic coating. We utilized the commercially available mr-DWL5 resist from Micro Resist Technology GmbH, Berlin, Germany. This resist is an epoxy-based negative resist designed for high aspect ratio lithography of thick resists with high sensitivity within the range of 350 to 410 nm. It was applied via spin coating onto a 100 mm Borofloat®33 glass wafer from Schott Technical Glass Solutions GmbH, Jena, Germany. The spin coating process consisted of 3400 and 1250 rpm for 30 s with a consistent acceleration of 500 rpm/s, aiming to achieve resist films of 5 μm thickness for compression channels and 12 μm for control channels, respectively. Prior to spin coating, the glass wafer underwent cleaning in an ultrasonic bath using acetone and isopropanol (IPA) for 3 and 2 minutes respectively, to eliminate particles, followed by drying with pressurized nitrogen flow. Subsequently, it was dehydrated at 200 °C on a hotplate for 10 min. To enhance adhesion to the photoresist, a commercial adhesion promoter (TI prime, MicroChemicals GmbH, Ulm, Germany) was spin-coated at 4000 rpm for 60 s. The adhesion promoter was activated through baking at 120 °C for 2 min, after which the mr-DWL5 resist coating was applied once the substrate cooled to room temperature. The resist film underwent soft baking, 50 °C for 5 min, 50 to 90 °C for 16 min (2.5 °C/min), and 90 °C for 40 min, on a programmable hot plate (PZ 28-3TD, Harry Gestigkeit GmbH, Germany), followed by natural cooling to room

temperature on the hot plate. Subsequently, the resist was allowed to relax at room temperature under ambient clean room conditions (approximately 21 °C, humidity 40–60%) for several hours before proceeding with the subsequent processes. Following exposure, the crosslinking process was initiated during a post-exposure bake, 50 °C for 5 min, 50 to 90 °C for 32 min (1.25 °C/min), 90 °C for 40 min, followed by natural cooling to room temperature. The resist was then developed in mr-DEV 600 (micro resist technology GmbH, Berlin, Germany) for a development time of 8 min, with gentle agitation, rinsed in IPA for 1 min, and dried with pressurized nitrogen flow. No hard-bake was conducted for complete crosslinking.

**Soft-lithography.** The DWL5 photoresist structure on the glass wafer was used as a master mold for fabricating a polydimethylsiloxane (PDMS) replica. To generate an anti-sticking coating on the wafer, an established controlled co-evaporation of silanes deposition was performed before PDMS casting[36]. The standard PDMS casting procedure involved mixing the PDMS base with its curing agent at a 10:1 ratio, pouring the mixture over the master structure, and curing it at 80 °C for 2 h. Once cured, the PDMS layer was removed from the mold, forming the microfluidic channel. After punching inlet and outlet holes, the PDMS and 24 mm × 50 mm × 0.17 mm ClariTex coverslips (CellPath Ltd) were plasma-treated, assembled, and sealed. A microfluidic system incorporating a neMESYS Low-Pressure syringe pump system (Cetoni GmbH) was used to fill the channels. To prevent nonspecific binding inside the microfluidic chip, a 1% BSA solution in PBS was pumped, followed by a PBS wash. Finally, the cell-containing solution was gently injected into the channel. Flow rates were set at 0.1 µl.s$^{-1}$ for the 1% BSA and PBS solutions.

**Microchannel compression of live PBMCs.** Live-stained PBMcs were subjected to compression in a 5 µm microchannel precoated with BSA. Specifically, 1 µl of cell solution consisting of RPMI 1640 medium and 20% FBS was injected into the channel at a flow rate of 0.005 µl.s$^{-1}$. Immediately afterward, the microchannel was placed on a microscope, and the channel was scanned, and the compressed cells were imaged over the total duration of 15 min using 405 nm and 599 nm channels. The Z-stack imaging of the cells was set with a step size of 1 µ, and the *xy* pixel size was set to 250 nm.

**Staining and live cell microscopy of PBMCs.** To prepare live PBMCs for fluorescence microscopy, the frozen cells were thawed in a 37 °C water bath and immediately diluted in PBS containing 2% FBS at a 1:4 ratio. After removing the cryopreservation media, the PBMCs were resuspended in PBS with 2% FBS, stained with NucBlue Live ReadyProbes™ Reagent by adding 800 µl.ml$^{-1}$ (Hoechst 33342, catalog no. R37605, Thermo Fisher) and DRAQ7 marker in a 1:200 ratio (Biolegend), incubated for 20 min at room temperature, washed three times with PBS with 2% FBS, and then used for microfluidic experiments. Confocal microscopy of the PBMC well-plate was performed using a Leica Stellaris (inverted DMI8) with LAS X software, using a 63X/1.4 NA oil immersion objective.

**Immunofluorescence staining.** To study DNA damage and the mechanical integrity of the nuclear lamina, a 96-well glass-bottom plate was coated with charged poly-L-lysine for 2 h at room temperature to promote cellular attachment. The poly-L-lysine was then removed, and the plate was washed with PBS until dry. Next, $2.5 \times 10^5$ cells fixed using PFA were plated onto the coated well plate for 6 h. Following the removal of PBS, the cells were permeabilized with 0.5% Triton X-100 for 10 min and washed twice with PBS. The cells were then blocked with 1% Bovine Serum Albumin (BSA) for 1 h. After removing the blocking solution, primary antibodies in the blocking solution (1% BSA in PBS-T) were added and incubated overnight at 4 °C. For immunostaining of PBMCs, Monoclonal mouse anti-Lamin A, dilution:1:200 (catalog no. ab8980, Abcam); Rabbit Polyclonal Anti-Histone H3 trimethylation K9, dilution:1:400 (catalog no. ab8898, Abcam); Mouse Monoclonal IL2RA Antibody, dilution:1:200 (catalog no. AB6411, Abcam) was used. After removing primary antibodies, the cells were washed using PBS-T for 10 min, twice, with gentle shaking. The following secondary antibodies were used: Alexa Fluor 555 anti-rabbit, dilution:1:1000 (catalog no. A32794) and Alexa Fluor 488 anti-mouse, dilution:1:1000 (catalog no. A32723) in 1% BSA (in PBS-T) for 2 h at room temperature in the dark. Following the incubation, secondary antibodies were removed, and the cells were washed with PBS-T for 15 min on a shaker. Finally, NucBlue Live ReadyProbes Reagent (Hoechst 33342, catalog no. R37605, Thermo Fisher) was added and incubated for 20 min, followed by a single wash with PBS. The cells were then left in PBS and imaged in PBS.

### Computational methods
**Nuclear segmentation.** All the images were of 8-bit depth, and the minimum and maximum values lie between 0 and 255. The images from different patient samples had differences in background and foreground intensity distribution, texture of the background, and minimum and maximum intensity values. Therefore, to segment the individual nuclei consistently across different patient samples from the background, we first employed Multi-Otsu thresholding with 3 classes to obtain 2 threshold values. We then conditionally selected the minimum of the two threshold values, if it was greater than 10; else we chose the maximum of the two threshold values. We then assigned unique labels to all the segmented regions by defining the connectivity of the different regions as 6. We ignored all the labeled regions with effective diameters of less than 4.5 microns and greater than 13.5 microns.

**Quantitative chromatin state representation using VAE and hand-crafted features.** The 3D segmented individual nuclear images ($N = 48{,}027$) of live PBMCs were maximum intensity Z projected and padded to make $128 \times 128$ pixel 2D fields of views (FOVs). The imaging data, consisting of 2D nuclear FOVs ($N = 48{,}027$), was divided into training (85.5%), validation (4.55%), and testing (10%) datasets. We then used a Variational Autoencoder (VAE) architecture as reported previously[37].

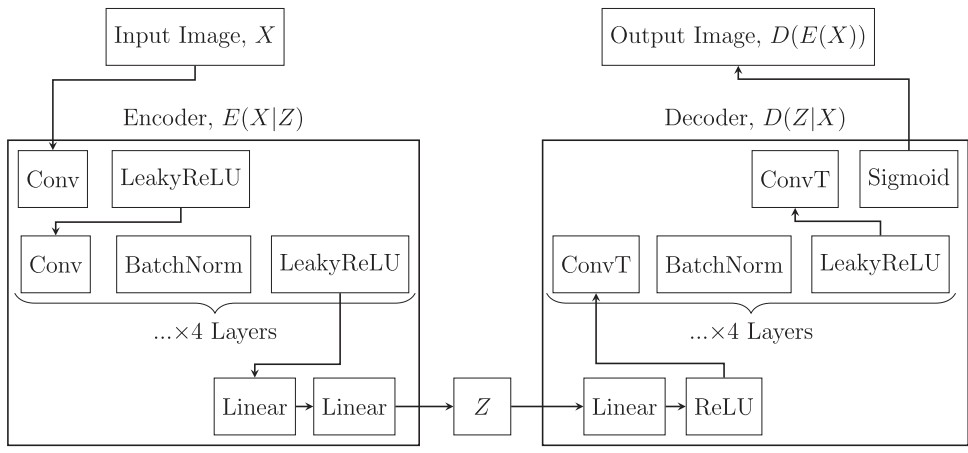

Briefly, as shown in the schematic above, the VAE consists of the encoder and decoder with 4 layers of cascaded convolution, batch normalization, and activation. The quantitative chromatin descriptor $Z$ is learned by jointly training an encoder and decoder function, both parameterized using neural networks, to minimize the loss function

$$\min_{E,D} E_{x \sim X} ||D(E(X)) - x||_2^2 + \lambda D_{KL}(p_E(.|x)|p_g(.)).$$

The first term quantifies the pixel-wise reconstruction error between the output and the input image, while the second term represents the Kullback–Leibler (KL) divergence between the latent space feature distribution, $p_E(.|x)$, and the Gaussian distribution, $p_g(.)$. Incorporating the second term into the loss function, as previously described in ref. [38], enforces the latent data distribution to approximate an isotropic Gaussian. During the training epochs, the training dataset was switched between individual disease conditions, i.e., Healthy, Prediabetic, and Diabetic, after every 20 epochs. The above model was trained until the reconstruction loss on images from the validation dataset is less than $1 \times 10^{-5}$, achieved after 600 epochs. After this, the quantitative chromatin descriptor, $Z$ is calculated for the training, testing, and validation datasets. The model was implemented using *Pytorch (version 1.13.1)* and trained using NVIDIA *Quadro RTX 4000* GPU, with a total training time of 16 h. The feature list used for calculating chromatin and nuclear morphometric features is the same as was described previously[39].

**Data sub-sampling and class balancing**. To optimally represent the number of nuclei from each disease condition and resolve the extreme imbalance in the total nuclei count from an individual in a particular disease condition, we used the following algorithm:

1. **Calculate mean count of nuclei ($\widehat{p}$)**:

For a given disease condition $d$, compute $\widehat{p}$, the mean count of nuclei across all individuals $i$ in that condition.

2. **Initial balanced dataset ($b_{i,d}$)**:

For each individual $i$ in the disease condition $d$, set the initial balanced count $b_{i,d}$ using the rule:

$$b_{i,d} = \begin{cases} \widehat{p}, & \text{if } p_{i,d} > \widehat{p} \\ p_{i,d}, & \text{otherwise} \end{cases}$$

This ensures that counts exceeding the mean are capped at $\widehat{p}$, creating a coarse balancing.

3. **Find the minimum nuclei count across conditions ($K_{\min}$)**:

For each disease condition $d$, sum the values of $b_{i,d}$ across all individuals $i$. Identify the minimum total count across conditions:

$$K_{\min} = \min\left(\sum_{i=1}^{n} b_{i,d}\right)$$

This will determine the target total count $K_{\min}$ to achieve balance across conditions.

4. **Create the well-represented dataset ($B_{i,d}$)**:

For each disease condition $d$, modify $b_{i,d}$ to construct $B_{i,d}$ by adjusting individual counts as follows:

$$B_{i,d} = \begin{cases} b_{i,d} - t, & \text{if } b_{i,d} = \max(b_{i,d}) \\ b_{i,d}, & \text{otherwise} \end{cases}$$

Here, $t$ is the number of nuclei removed iteratively from the highest $b_{i,d}$ values until the total nuclei count in $B_{i,d}$ matches $K_{\min}$:

$$\sum_{i=1}^{n} B_{i,d} = K_{\min}.$$

**Identification of distinct clusters of chromatin states**. To identify distinct chromatin states in the dataset, we followed these steps:

1. **Consensus Leiden Clustering**:

To obtain robust cluster label assignments, we applied a consensus clustering strategy[40] using the Leiden algorithm[41] as the base partitioner. First, a k-nearest neighbor (kNN) graph was constructed from the cell-feature matrix using Euclidean distance and a fixed neighborhood size ($k = 10$, unless stated otherwise). We then performed $n = 30$ independent Leiden runs with varying random seeds at a resolution parameter of 1.0. For each run, the cluster assignments were recorded, and a co-association matrix was built, where entry $(i,j)$ reflects the proportion of runs in which cells i and j were assigned to the same cluster.

The co-association matrix was subsequently converted into a dissimilarity matrix ($D = 1-P$), which was subjected to average-linkage hierarchical clustering. The final consensus partition was derived by cutting the dendrogram at a threshold corresponding to a minimum co-association of 0.8, thereby retaining only densely supported clusters.

To assess robustness, we computed the Adjusted Rand Index (ARI) between the consensus partition and each individual Leiden run. Reported stability scores include the mean and standard deviation of ARI across runs. This procedure ensures that the reported clusters represent stable and reproducible communities, rather than artifacts of stochastic initialization. We iteratively run the above algorithm by removing the clusters with fewer than 30 cells until the sizes and number of clusters converge, and/ or we obtain the mean ARI of ≥0.8.

2. **Calculating disease-specific enrichment**:

For each disease condition $d$ and cluster ID $h$, we computed the number of cells $K_{d,h}$ in each cluster $C_{z_h}$, where $h \in [0, 9]$(including UC). To assess the disease-specific enrichment of each cluster, we normalized the count $K_{d,h}$ by $K_{\min}$, the total cell count for the disease condition $d$, as follows:

$$\text{Enrichment}_d = \frac{K_{d,h}}{K_{\min}}$$

where Enrichment$_d$ represents the relative enrichment of cluster $C_z$ for disease stage $d$.

3. **Calculating cluster-specific enrichment**:

For each disease stage $d$ and cluster ID $h$, we computed the number of cells $K_{d,h}$ in each cluster $C_{z_h}$. To assess the enrichment profile of every cluster for the disease stage, we normalized the count $K_{d,h}$ by $K_h$, the total cell count in cluster $h$, as follows:

$$\text{Enrichment}_h = \frac{K_{d,h}}{K_h}$$

where Enrichment$_h$ represents the relative enrichment of cluster $C_z$ for disease stage $d$.

**Individual specific cluster enrichment and prediction**. For each individual $i$ and cluster ID $h$, we computed the number of cells $K_{i,h}$ in each cluster $C_{z_h}$. To assess the enrichment profile of every individual across clusters, we normalized the count $K_{i,h}$ by $K_i$, the total cell count of the individual $i$, as follows:

$$\text{Enrichment}_i = \frac{K_{i,h}}{K_i}$$

where Enrichment$_i$ represents the relative enrichment of individual $C_z$ in different cluster $h$. We then trained a Random forest classifier (RFC), with 10 estimators, on the enrichment profile from all the individuals except the held-out one as input and the corresponding disease annotation as the label. We then predicted the disease based on the enrichment profile for the held-out individual as input. For patient-specific prediction, we removed all patients with less than 100 cell counts.

**Getting individual specific cluster enrichment on independent set.** We transferred cluster labels from a training split to an independent testing set using kNN voting in the reference feature space. A k-NN index ($k = 10$, cosine distance) was fit on the reference features. For each query sample, we retrieved its $k$ nearest reference neighbors, tallied their labels, and used the normalized vote proportions as per-label probabilities; the predicted label is the arg max of these probabilities. Assignment confidence combines (i) the vote probability $p$ for the predicted label and (ii) a distance-based compactness term. Let $d$ be the query sample's mean distance to its $k$ neighbors. We estimate $\mu_{\mathrm{ref}}$ and $\sigma_{\mathrm{ref}}$ as the mean and standard deviation of mean neighbor distances computed on the reference set (excluding self-matches). Confidence is

$$\text{confidence} = \frac{1}{2} p + \frac{1}{2} \sigma\left(\frac{\mu_{\mathrm{ref}} - d}{\sigma_{\mathrm{ref}}}\right), \ \sigma(x) = \frac{1}{1 + e^{-x}}.$$

Predictions with confidence below a tunable threshold ($t$) were rejected. The output reports the predicted label, vote probability, confidence, mean neighbor distance, a rejection flag, and per-label probabilities.

**Immunofluorescence imaging data quantification.** The protein expression for nuclear stains such as Lamin and H3K9Me3 is evaluated for individual nuclei on the labeled mask obtained after nuclear segmentation. The protein expression value reported for the nuclear stains in the study is the summation of the total voxel-wise intensity of protein evaluated over the mask, divided by the total volume of the mask. The expression value of the cytoplasm or cell-membrane specific proteins, such as CD25 is evaluated by dilating the nuclear mask by 2 μm from the nuclear labels obtained from nuclear segmentation. The value of cytoplasm or cell-membrane specific protein expression is then min-max scaled, and the normalized expression value is evaluated by summation of the total voxel-wise intensity of protein intensity inside the mask, divided by the total volume of the dilated nuclear mask. The threshold is calculated on the histogram of the normalized value of nuclear intensity from all nuclei from different disease stages and is evaluated using the Otsu Threshold. Cells with protein expression above the threshold are considered positive for CD25 expression as shown in Fig. S12A.

**Statistics and reproducibility.** All statistical analyzes and plotting were carried out in Python[42]. For box plots, the box limit represents the 25th to 75th percentile, and whiskers represent 1.5 × interquartile range. We evaluated the statistical significance of the mean with the Mann–Whitney U-test, performed between a sample of interest and the corresponding control. *$P < 0.05$; **$P < 0.01$; ***$P < 0.001$; ****$P < 0.0001$. Statistical data visualization and representation were carried out using Scikit[43], Matplotlib[44], and Seaborn[45] library.

# Results
## Chromatin imaging of live PBMCs in patients with Type 2 Diabetes Mellitus (T2DM)
We selected a cohort of 57 individuals comprising healthy donors and patients with prediabetes and diabetes with ages ranging from 25 to 80 years, as shown in Fig. 1A. Given that T2DM exhibits sex-specific differences in comorbidities and complications[46], our cohort included ~40% female and ~60% male participants, distributed across various age groups and disease stages (Fig. 1B). For this study, we obtained whole blood liquid biopsies from the patients visiting the Hausarztpraxis MZ Brugg, Switzerland. In Fig. 1C

and Supplementary Table S1, we show the primary disease diagnosis and concurrent diseases of the cohort of patients at different stages of T2DM. Hyperlipidemia, with 10 and 12 recorded instances, and Hypertension, with 9 and 14 instances, are identified as the most frequently co-occurring conditions throughout the stages of T2DM in our cohort. We used the standard gradient separation method to isolate the live PBMCs as shown in Fig. 1D, and "Materials and Methods". We transferred the live PBMCs stained with Hoechst and DRAQ7 onto the chromatin imaging micro-channel device, with a channel depth of 12 μm (Fig. 1D, "Materials and Methods"). We captured the volumetric chromatin distribution of live PBMCs loaded in microchannel confinement using a confocal microscope, as shown by 2D projections in *XY* and *XZ* planes in Fig. 1E and "Methods". We segmented the 3D nuclear volume using the algorithm detailed in Fig. S1A and the "Materials and Methods" section. In the Supplementary Table S2, we show the total number of segmented nuclei for each disease stage. Representative z-projected confocal images of live PBMC nuclei stained with Hoechst from each individual, along with their corresponding segmentations, are presented in Figs. S1B, S2A, S3A, S4A, and S5A.

## PBMC subpopulations identified through embedding live chromatin images distinguish the stages of T2DM
Using a microfluidic chromatin imaging assay, we obtained 3D confocal images of Hoechst-stained live PBMC nuclei from healthy, prediabetic, and diabetic individuals, which exhibited variability in nuclear count (Fig. S6A). To reduce this sample size-dependent variability, we downsampled our dataset for subsequent analysis, ensuring a well-represented dataset of nuclei across all disease stages and individuals (see "Materials and Methods"). The final downsampled nuclei count for each individual across different stages of T2DM is shown in Fig S6A. Despite the variability in nuclear sizes within and among patients (Fig. S6A), we observed that PBMCs from diabetic individuals exhibited larger nuclear sizes compared to those from pre-diabetic and healthy individuals (Fig. S6B, C). To further investigate changes in nuclear morphology and chromatin organization during T2DM progression, we applied to the sub-sampled chromatin images representation learning techniques that have previously been shown to predict gene expression profiles and disease progression across various diseases[35,37]. We trained a VAE to compress and reconstruct the 2D maximum z-projections of the segmented 3D confocal nuclear images from the PBMCs of each individual (Fig. 2A), following the architecture as described in previous studies[35,37,38]. We extracted a compact set of latent features that capture key structural and morphological characteristics of every nucleus through the VAE and clustered the cells using graph-based neighborhoods, as shown in Fig. 2A and explained in "Materials and Methods". Exhaustive VAE embedding of nuclear images from individual patients and from healthy, prediabetes, and diabetes after sub-sampling is presented in Figs. S6D, S7A, S7B, S8A, and S8B.

Consensus Leiden Clustering grouped the nuclei into 9 distinct chromatin clusters, CVAE, describing specific PBMC subpopulations as shown in Fig. 2B (see "Methods"). It can be seen that the PBMC sub-clusters with similar physical characteristics, for example, clusters 8 and 9 with visibly larger projected areas and clusters 1 and 2 with smaller projected nuclear areas in Fig. 2B, lie closer in the reduced-dimensional embedding (Fig. 2B, C). Randomly splitting the cells into train and test sets, we trained a RFC to distinguish between the 3 disease stages (healthy, prediabetic, diabetic) based on the handcrafted and VAE-derived features listed in the SI and our previous work[39]. We evaluated the most important features for disease stage discrimination by GINI importance score, as shown in Fig. S9B. Among the handcrafted descriptors, we find that the heterochromatin organization features describing the radial location of heterochromatin domains from the center and separation of heterochromatin domains from the nuclear boundary are found to be strongly associated with pathology class (Fig. S9C). The embedded features obtained from the VAE also capture these key chromatin organization features, as shown in Fig. 2D, E. We find that we can distinguish the different disease stages with a high degree of accuracy on cells held out from training (Figs. 2F and S9A). Notably, we find

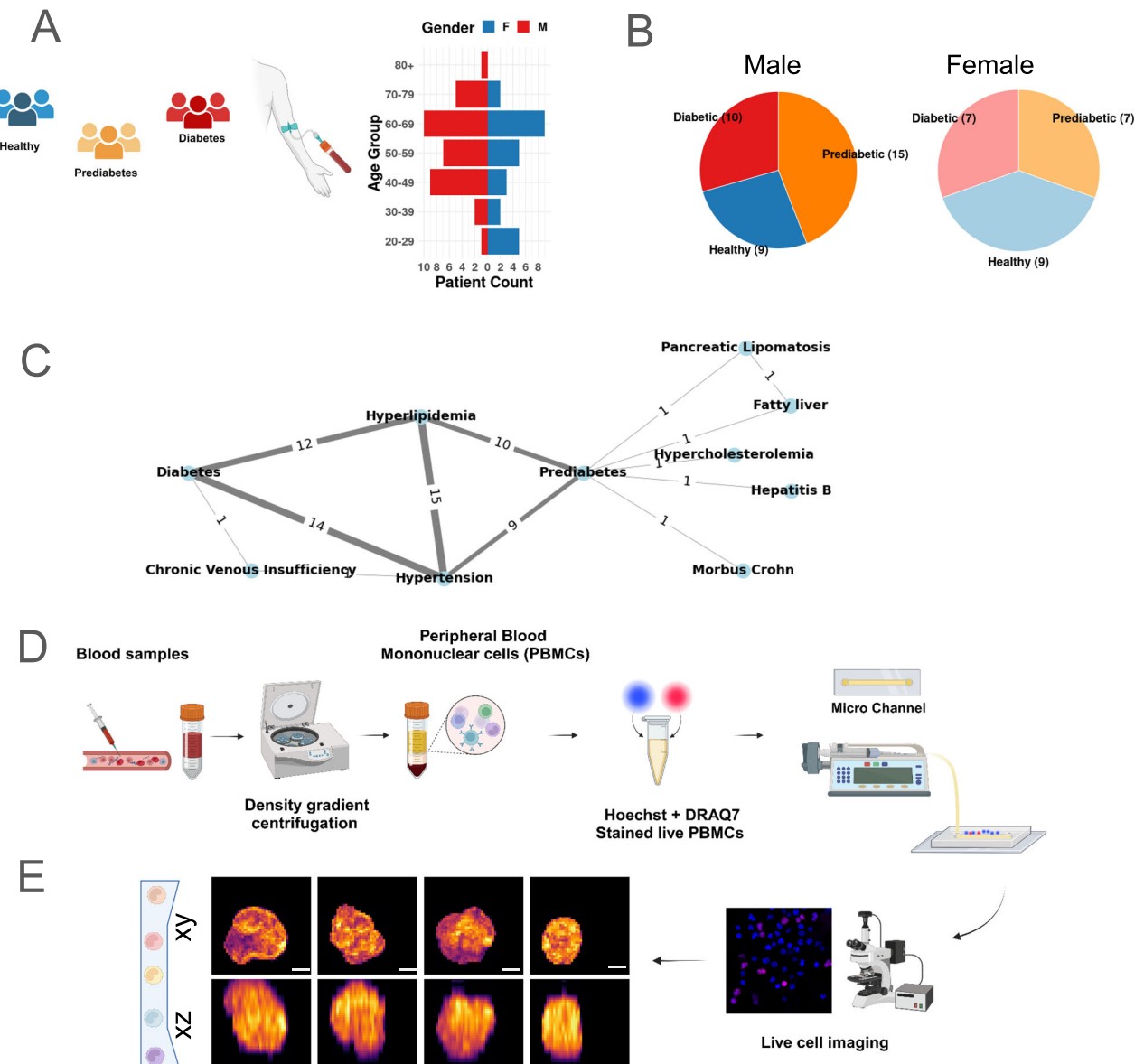

**Fig. 1 | Schematics showing study design as well as compression and imaging assay of live PBMCs. A** The study cohort of 57 individuals at different stages of T2DM. Blood samples obtained from these individuals were analyzed to identify adaptive immune responses to T2DM. **B** Pie charts showing the number of individual samples from each disease category corresponding to their respective genders. **C** Cohort disease co-occurrence graph, with node annotations representing primary diagnosis of cohort and edge annotations representing instances of co-occurrence of corresponding diseases. **D** Schematic of live PBMC imaging assay. PBMCs isolated from the blood samples were stained with Hoechst and DRAQ7. Live PBMCs were loaded in a microchannel with a height of 12 μm and imaged using confocal fluorescence microscopy. **E** Representative images of *XY-XZ* projection of PBMCs stained with Hoechst showing chromatin organization. Scale Bar: **C** 2 μm.

that $C_{VAE}$ clusters 6, 7, and 8 are particularly enriched with PBMC chromatin images from diabetic samples, cluster 1 is enriched with nuclei from prediabetic samples, and cluster 4 is enriched with nuclei from healthy samples (Fig. 2G). The identified clusters were robust to variations in initial conditions, achieving an ARI of 0.85 ± 0.10 (Fig. S9D) after excluding nuclei that did not cluster. These nuclei, localized outside of the 9 distinct $C_{VAE}$ clusters, henceforth labeled as "UC", accounted for 7.07% of total cells.

Motivated by assigning interpretable, quantitative descriptors of alterations to chromatin state, we performed consensus Leiden clustering, this time using our handcrafted features instead of the VAE features used above. With the handcrafted features, we derive 15 distinct $C_{HFE}$ clusters as shown in Fig. S9F, with $C_{HFE}$ clusters 4, 11, 12, 13, 15 and $C_{HFE}$ clusters 0, 2, 5, 6, 8 consisting of a higher fraction of nuclei belonging to prediabetic and diabetic individuals. The $C_{HFE}$ cluster achieved a remarkably high ARI of 0.91 ± 0.01, agnostic to initial conditions (Fig. S9G, H). We find that there is

some association in cluster identities assigned by these two parallel methodologies, indicated by the highest (10% to 20%) fraction of nuclei sharing the same cluster labels in $C_{VAE}$ and in $C_{HFE}$, as shown in Fig. S9I. Moreover, individual clusters derived from VAE ($C_{VAE}$) or handcrafted ($C_{HFE}$) embeddings enrich nuclei with different morphometric and chromatin organization characteristics, as shown in Fig. S9J, K. Taken together, these results suggest that the subcluster enrichment of feature-based embeddings of chromatin images can be used to differentiate between the healthy, prediabetic, and diabetic individuals across our cohort.

**PBMC subpopulations identified from embedded chromatin images predict patient-level changes in chromatin states in T2DM**

Given the subpopulation-specific enrichment observed at various stages of T2DM, we aimed to utilize the 10-dimensional $C_{VAE}$ (including UC)

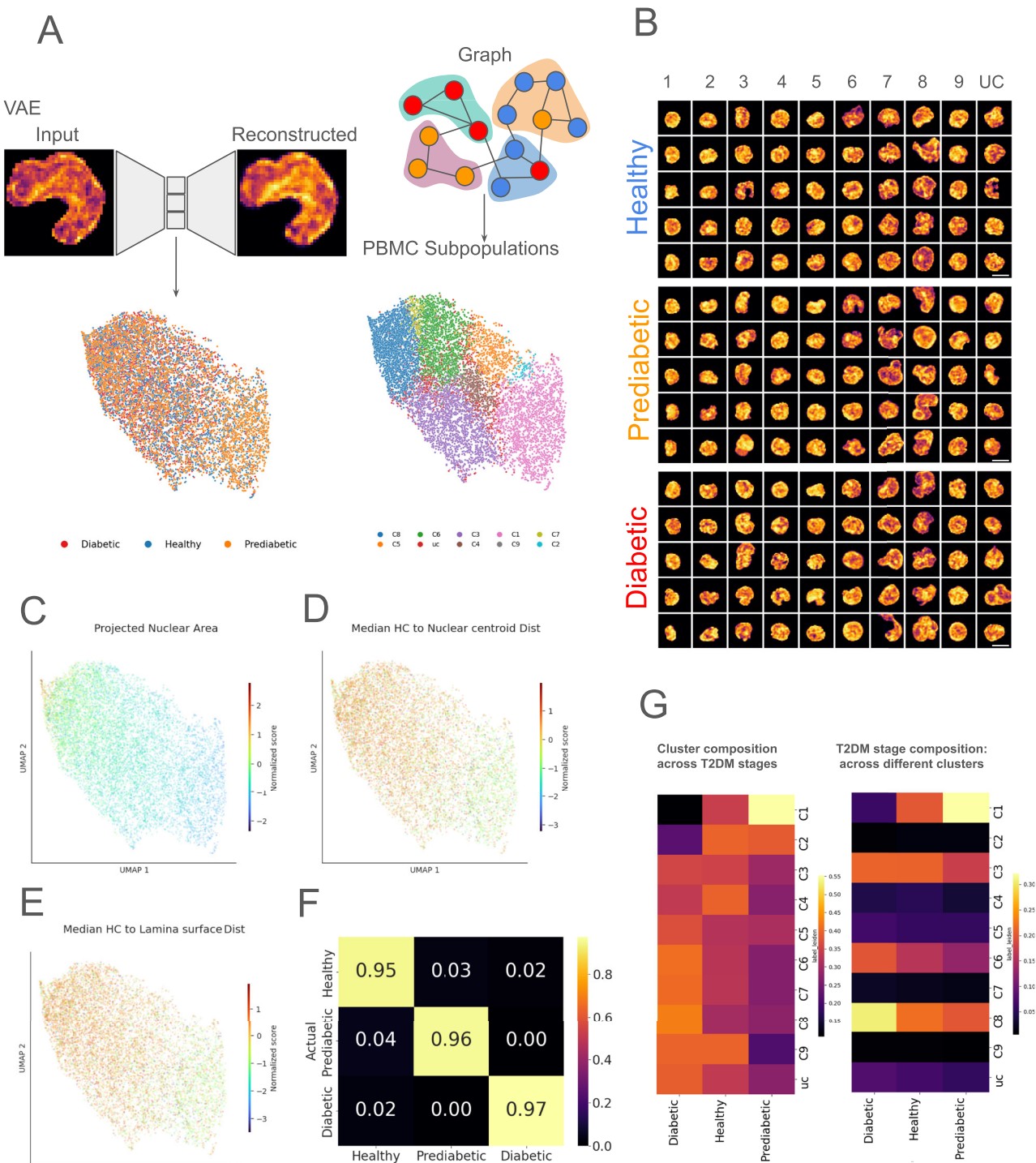

**Fig. 2 | PBMC chromatin states at different stages of T2DM. A** Latent space embedding derived from variational autoencoder reconstruction, followed by Leiden clustering on the K nearest neighbor graph. **B** Representative images of the nucleus of PBMCs from each cluster. **C–E** Showing the overlay of 2D projected area of nuclei, median dense chromatin region to nuclear lamina separation, median dense chromatin region to nuclear centroid separation across the embedding space. **F** Classification accuracy of prediction on cells held-out from training based on hand-crafted features. **G** (left) Cluster-specific enrichment profile, $Enrichment_h$ (see "Methods"), representing the fraction of cells from each T2DM disease stage in a cluster. Warmer colors represent higher fraction of a particular pathology in a cluster. Row sums to 1. (right) Disease-specific enrichment profile, $Enrichment_d$ (see "Methods"), representing fraction of cells from a different clusters in a disease stage (colder colors across all columns (for e.g., 9 and 2) representing less populous and smaller clusters). Columns sums to 1. Scale Bar: **D** 5 μm.

and 16-dimensional $C_{HFE}$ cluster enrichment profiles of individual patients to make predictions about their pathological outcome. Due to the low dimensionality of the enrichment profiles and the limited number of patients, we employed a RFC trained on these profiles to predict the T2DM disease stage for each patient (Fig. 3A). We evaluated individual patients' enrichment profiles across healthy, prediabetes, and diabetes from $C_{VAE}$ and $C_{HFE}$, and used them for different comparisons shown in Fig. S10A–D and Fig. S11A–D, respectively. Across different comparisons, the balanced accuracy of our model in 2-way classification tasks – distinguishing between diabetic vs. healthy—is 0.58 from $C_{VAE}$ and 0.72

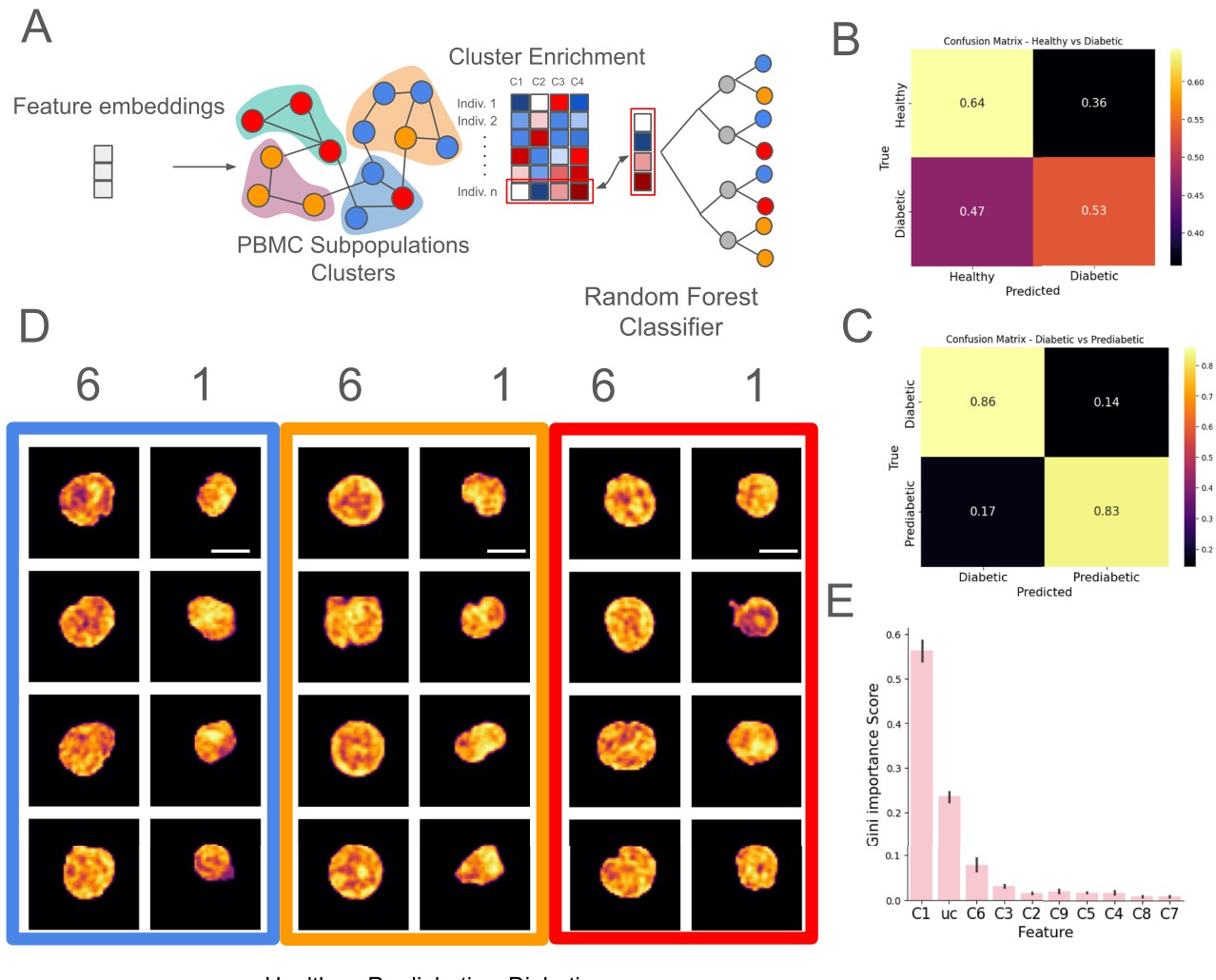

**Fig. 3 | Patient-specific prediction of immune cell alterations in T2DM.**
**A** Schematic showing prediction task using cluster enrichment vector for individual patients. Confusion matrix for leave-one-patient-out prediction for **B** Healthy vs Diabetic, and **C** Prediabetic vs Diabetic. **D** Representative images of cells from Cluster # 1 and # 6 from each of the disease types. **E** Feature importance scores show the cluster type responsible for the prediction of disease type in each patient (one-way ANOVA $p$-value: 1.215E–100, Error Bar: 95% Confidence Interval, $n_{Diab}$ = 14, $n_{Healthy}$ = 18, $n_{PreDiab}$ = 21). Scale Bar: **D** 5 μm.

from $C_{HFE}$ with AROC of 0.65 as shown in Fig. 3B and Fig. S12A, D validated against the null hypothesis by permutation testing as shown in Fig. S12G. Between diabetic vs. prediabetic individuals held-out from training we find balanced accuracy of 84% with AROC of 0.70 from $C_{VAE}$ and AROC of 0.85 with $C_{HFE}$ as shown in Figs. 3C and Fig. S12B, E validated against the null hypothesis by permutation testing as shown in Fig. S12H. The classification results of Healthy vs Prediabetic between the two approaches differ markedly; we see increased balanced accuracy of 0.7 and AROC of 0.77 with $C_{HFE}$ against the null hypothesis, as shown in Fig. S12C, F, I compared to $C_{VAE}$ (Fig. S10G). Although chromatin images from individual T2DM stages looked characteristically similar (Fig. 2B), cluster $C_{VAE}$ 1 was significantly enriched in prediabetic patients (Fig. S10K), while cluster $C_{VAE}$ 6 showed significant enrichment in diabetic individuals (Fig. S10L). This was confirmed by the mean cumulative cluster importance score as shown in Fig. 3E and validated by the mean GINI importance score in Fig. S12J.

We derive similar but stronger enrichments using the $C_{HFE}$ cluster, in particular for clusters $C_{HFE}$ 4, 11, and 13, where we see significant enrichment of nuclei from individuals with prediabetes. Within cluster $C_{HFE}$ 5, nuclei from diabetic individuals showed a significantly greater enrichment than healthy individuals, relative to the prediabetic individuals that showed minimal enrichment. Notably, $C_{HFE}$ 5 enriches for nuclei with maximum

radial location of heterochromatin domains from the center "D3" and maximum separation of heterochromatin regions from the nuclear boundary represented by annotation "D4", shown in Fig. S9J. The three-way prediction between healthy, diabetic, and prediabetic individuals for both cluster enrichment features, we see higher recall and one vs rest AROC for prediabetic and diabetic individuals, as shown in Figs. S10E–H, and S11E–H.

Next, we estimate the performance of our method on the general cohort, using independent set analysis. To this end, we randomly split the training and testing sets before preprocessing into two separate cohorts of individuals, as shown in Fig. S13A, B. We then cluster and evaluate the enrichment vectors for training data as described previously. For testing data, we assign the cluster labels to the data using the KNN distance-based majority voting with training data as a reference to evaluate the enrichment vectors (see "Methods"). Training on enrichment vectors for testing the data, we evaluated the prediction accuracy for the independent testing data. We find that the performance of the model is in line with the results observed in this study, with balanced accuracy of 75% and 77.5% for Pre-diabetic vs Diabetic & Healthy vs Diabetic individuals, respectively, at the default thresholds as shown in Fig. S13D, F. However, we saw significant confounding between Healthy and Prediabetic individuals, as shown in Fig. S13G. With a target sensitivity of 0.8 for Diabetic and Prediabetic

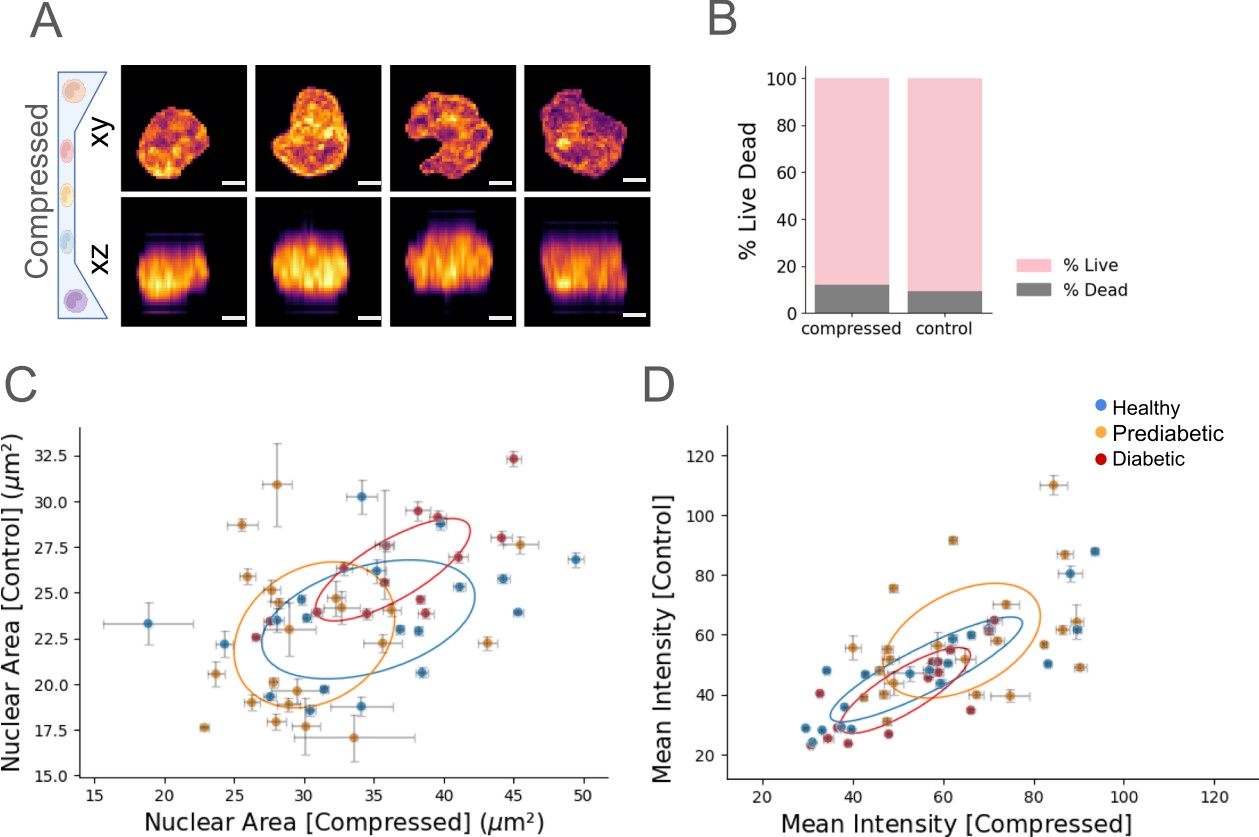

**Fig. 4 | Mechanical characterization of PBMCs in T2DM. A** Representative images of XY-XZ projection of live PBMCs stained with Hoechst under compression showing chromatin reorganization. **B** Characterization of viability of live PBMCs under compression compared with control using DRAQ7 intensity. Scatter plots showing (**C**) projected area (Hotelling's *T*-Test *p*-values: Healthy vs. Diabetes: 0.053029, Healthy vs. Prediabetes: 0.200034, Diabetes vs. Prediabetes: 0.006393, $n_{Diab} = 14$, $n_{Healthy} = 18$, $n_{PreDiab} = 21$) and (**D**) mean Hoechst intensity (Hotelling's *T*-Test *p*-values: Healthy vs. Diabetes: 0.396768, Healthy vs. Prediabetes: 0.261179,

Diabetes vs. Prediabetes: 0.024917, $n_{Diab} = 14$, $n_{Healthy} = 18$, $n_{PreDiab} = 21$) under compression against control for every patient in healthy, prediabetic and diabetic individuals. The *X* and *Y* error bars indicate the standard error in the projected area and mean Hoechst intensity for each patient in the control and compressed condition. The elliptical annotation indicates the region of one standard deviation from the mean. The error bar on the points indicates the standard error of mean (SEM). Scale Bar: **C** 2 μm.

individuals, and a target specificity of 0.8 for healthy individuals defined as the threshold for the training model, we find a specificity of 0.78 (CI: 0.4–0.972) and sensitivity of 0.75 (CI: 0.194–0.994) in distinguishing diabetic individuals for the independent set, as shown in the Supplementary Table S4 of the Supplementary Information.

These findings suggest that the quantitative representation of chromatin organization in PBMCs, using descriptive features, can provide clinically relevant predictions of chromatin alterations in individuals with T2DM.

### Mechanical and functional characterization of PBMCs during stages of T2DM

Immune cells undergo constant transmigration and egression during various immunological responses. During these processes, they are invariably compressed both at the endothelial barrier and in the interstitial tissue microenvironment, which alters their pliability. We therefore hypothesized that the mechanical properties of PBMCs can inform immune-related alterations during T2DM progression. To investigate this, we analyzed the alteration in mechanical properties in the nuclei of PBMCs in the respective stages of T2DM (Fig. 4A). We developed an in vitro assay to assess the mechanical properties of PBMCs using microchannel (5 μm) compression. Next, we performed live cell chromatin imaging of PBMCs from both healthy, prediabetic, and diabetic individuals under microchannel compression (Fig. 4A). We characterized the viability of PBMCs during imaging by triangle thresholding the DRAQ7 nuclear intensity and found 90% and

86% viability in PBMCs confined in control and compressed in the microchannel, respectively (Figs. 4B and S14A). We found chromatin condensation and flattening of PBMC nuclei as evidenced by the increase in mean intensity of Hoechst and increase in nuclear volume as well as projected area (Fig. S14B–D). Notably, we found marked changes in the nuclear deformation in diabetic individuals as shown in Fig. 4C. We also found greater chromatin condensation in prediabetic individuals, as shown in Fig. 4D.

To understand the functional basis of such organizational and mechanical alterations to chromatin in T2DM, we next performed a series of immunofluorescence experiments (Fig. 5A). In particular, we examined lamin expression, as our previous findings indicated that nuclear size and mechanical properties of PBMCs are significantly altered in T2DM (Figs. 2C, S6B, C and 4C). We found a significant decrease in Lamin A/C expression in diabetic individuals compared to healthy controls (Fig. 5B, C). We reasoned that the observed increase in nuclear size may, in part, be attributed to enhanced activation of immune cells, as noted in our previous study[47]. We find a heterogeneous increase in the fraction of immune-activated PBMCs from individuals with Diabetes compared to healthy controls, measured by the increased surface expression of IL-2 receptor α-chain (CD25) (Fig. 5D, E). The observed increase in nuclear size and altered mechanical properties prompted us to examine heterochromatin expression in PBMCs in T2DM. We saw a marginal increase in heterochromatin levels given by H3K9ME3 expression in individuals with T2DM compared to healthy

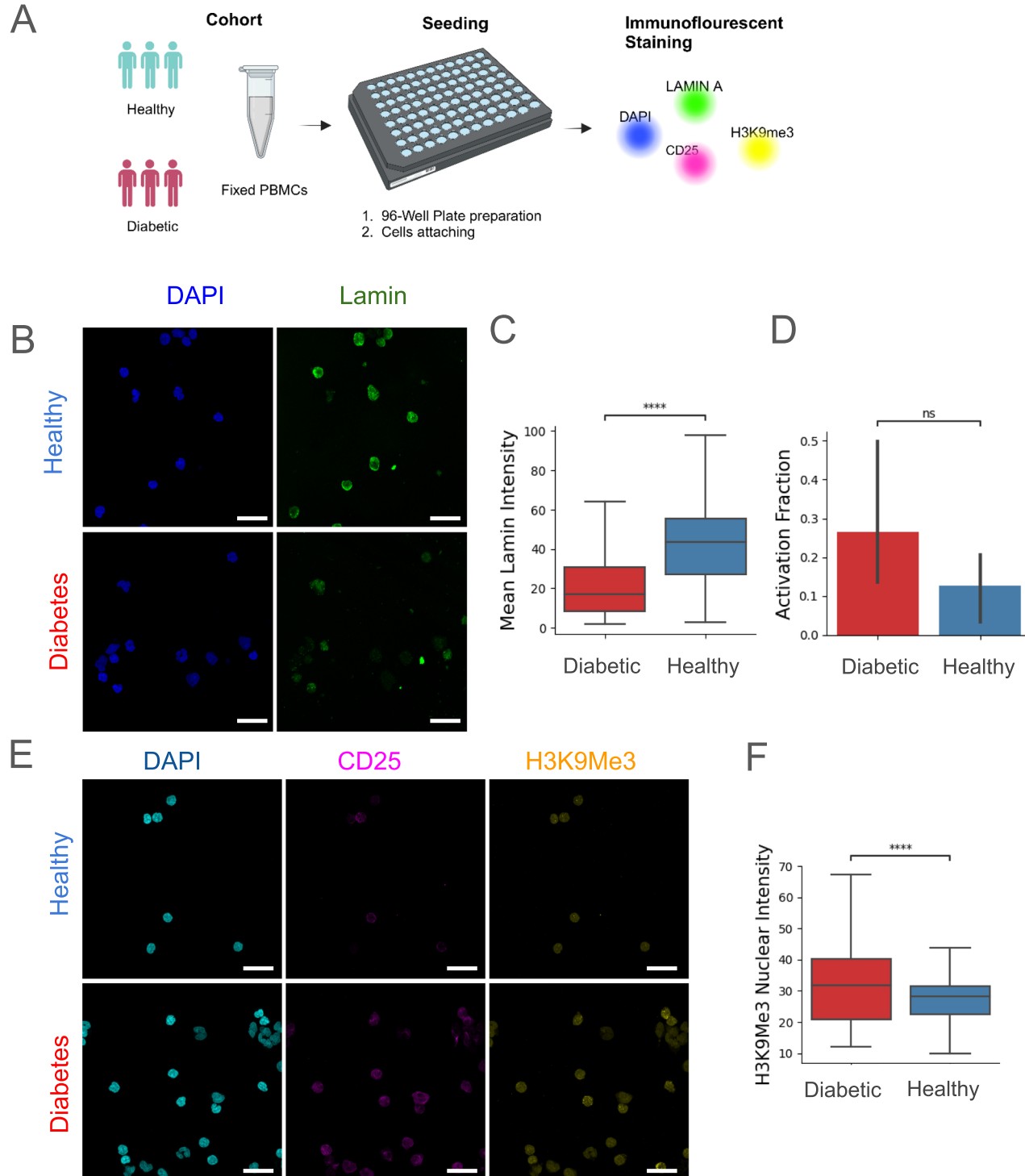

**Fig. 5 | Functional chromatin changes in PBMCs during T2DM. A** Schematic workflow of the experiments evaluating protein expression and functional state of PBMCs from selected healthy and diabetic individuals. **B** Representative images of PBMCs stained for Lamin and DAPI. **C** Box plot showing differences in Lamin intensity in healthy and diabetic individuals (# individuals = 3, Mann–Whitney U test $p_{val}$:1.282e−84)). **D** Stacked bar plot represents the labeled fraction of CD25+ cells in healthy and diabetic individuals (# individuals = 3, Mann–Whitney U test $p_{val}$: 7.000e−01). **E** Representative images of cells from healthy and diabetic individuals. XY and XZ projections for different protein labels, CD25, H3K9Me4, and DAPI. **F** Single-cell nuclear H3K9me3 intensity of PBMCs in healthy and diabetic individuals (# individuals = 3, Mann–Whitney U test $p_{val}$: 9.281e − 25). Scale Bar: **B** & **E** 20 μm, *p*-value obtained using Mann–Whitney U Test.

controls (Fig. 5F), and a higher expression of heterochromatin in PBMCs was associated with more activation (Fig. S15C). We did not find the expression of Lamin, H3K9Me3, and IL-2 receptor α-chain to be highly dependent on the morphology of the PBMC nuclei (Fig. S15D–F). These results reveal alterations in nuclear mechanics and chromatin condensation states of PBMCs in individuals with T2DM, further supporting that chromatin architecture could serve as a sensitive indicator of diabetes-associated immune dysfunction

## Discussion
In summary, we used liquid biopsies from 57 individuals at various stages of T2DM to develop a chromatin-based imaging biomarker capable of

predicting immune-related alterations during T2DM progression. By leveraging a convolutional autoencoder, we extracted image features that identify distinct chromatin states and nuclear morphologies, enabling us to distinguish between different stages of the disease. Our analysis revealed that specific immune cell subpopulations are enriched in patients with T2DM and healthy individuals, respectively. Quantitative descriptors capturing loci of condensed chromatin, nuclear size, and intensity distribution were the key features that enabled identification of the enriched PBMC subpopulation specific to diabetes and prediabetes. These observations align with previous studies on alterations in immune cell subpopulations in metabolic disorders[8,48] and demonstrate how live PBMC chromatin imaging can characterize functional changes in immune cells.

We further validated these findings using immunofluorescence, showing that PBMCs from patients with T2DM exhibit decreased Lamin expression and increased cellular activation. We also find an increase in heterochromatin and an association of activation with higher heterochromatin. These results support our analysis of Hoechst-stained images, which identified chromatin remodeling-associated features (e.g., condensed chromatin loci, nuclear size, and intensity distribution) as important in differentiating disease stages of T2DM. This is also in line with previous studies showing that in T2DM, hyperglycemia induces persistent epigenetic remodeling-such as altered H3K9me3 at inflammatory and regulatory loci in endothelial cells, monocytes, and progenitors-that silences anti-inflammatory brakes while sustaining NF-$\kappa$B activity, thereby favoring subset-specific (e.g., CD25+ T cell) rather than global PBMC activation[49–52]. Previous studies have shown that the chronic hyperglycemic and pro-inflammatory environment in T2DM can result in changes in the mechanical properties in different cell types during T2DM progression[47,53,54]. We employed a simple compressive loading device designed to replicate the physiological conditions to simulate the narrow constrictions that immune cells routinely traverse during transmigration across tissue barriers. We initially hypothesized that applying force would provide additional insights into the force-dependent chromatin states of immune cells in different patient cohorts. Although we did not observe a significant increase in detection accuracy, we did identify subtle changes in nuclear mechanical properties at all the stages of T2DM progression. These results suggest that nuclear mechanics may reflect the underlying cellular or molecular alterations associated with the disease. Since force-related nuclear deformation remains an active area of research, future studies could explore varying both the magnitude and duration of force application to enhance chromatin-based detection accuracy. Using the available covariates (age, sex), we tested associations between cluster-enrichment scores (derived from handcrafted features) and these covariates within and across diagnostic groups. We observed no meaningful correlation with age (Pearson's $r \leq 0.182$; Fig. S16A–Q). In addition, while there are some sex-related differences in the cluster enrichment scores of prediabetic and diabetic cohorts, these differences are not statistically significant after controlling for the false discovery rate using Benjamini–Hochberg correction (Fig. S17A–C).

Existing clinical biomarkers, such as FPG, OGTT, and glycated hemoglobin (HbA1c)[55] to detect and monitor T2DM, do not provide a clinically relevant readout of low-grade immune dysfunction observed in T2DM patients. Existing inflammatory biomarkers such as C-reactive protein (CRP), Tumor Necrosis Factor-$\alpha$ (TNF-$\alpha$), except Interleukin-6 (IL6), show significant heterogeneity across patients[56]. In this study, we have demonstrated the use of live chromatin imaging of PBMCs to provide a clinically relevant readout of immune-related alterations in patients with T2DM. When applied to our cohort, our approach achieved a specificity of 0.78 (CI: 0.4–0.972) and sensitivity of 0.75 (CI: 0.194–0.994) in distinguishing diabetic individuals. This is similar to currently available immune-related tests for detecting diabetes, including HbA1c, CRP, and IL-6. HbA1c with well-validated cut-offs (HbA1c ≥6.5%) achieves high specificity (≥90%), but moderate sensitivity (45–70%)[57–59]. CRP and IL-6 are established inflammatory markers associated with increased risk of T2DM (e.g., CRP relative risk 4.2, IL-6 relative risk 2.3)[60], and in multivariate models combined with other biomarkers (such as adiponectin, ferritin, interleukin-

2 receptor A (IL2RA), glucose, and insulin) achieve accuracies around 76–78% in predicting at risk patients[61] (Supplementary Table S3). From a cost perspective, HbA1c and CRP are low-cost assays[62,63], while IL-6 is substantially more expensive[64] (Supplementary Table S3). Our approach remains inexpensive and uses low-cost reagents such as Hoechst dye and microfluidics devices[65], suggesting that material costs could be comparable with scalability (Supplementary Table S3). Importantly, our approach provides complementary information about the functional state of the immune cell that is not directly addressed by established biomarkers. While single-cell resolution genomics approaches could also provide cell-specific functional readout, cost-effective, uncomplicated, and clinically scalable blood cell chromatin imaging provides unique possibilities.

Nevertheless, several limitations should be noted. For instance, our algorithms were developed and tested in a single cohort, and independent validation in larger, multi-center cohorts will be required to establish generalizability. The dependence on research lab infrastructure (e.g., centrifuge, confocal microscope, workflow complexity) can be addressed by developing specialized automated imaging and screening workflows. Moreover, our approach does not provide detailed immune cell subpopulation resolution, nor did we perform functional assays (e.g., cytokine secretion, T cell or macrophage activation studies) to mechanistically link chromatin architecture to immune dysfunction. Future studies integrating chromatin imaging with single-cell sequencing or proteomics will be critical to further validate and extend our findings.

## Data availability

All processed data is available in the main text or the supplementary materials. The raw imaging data along with the quantification tables is available at BioImage Archive with accession number S-BIAD2905 and https://doi.org/10.6019/S-BIAD2905.

## Code availability

The code is available at the following link: https://github.com/Rajshikhar/Detecting-chromatin-state-alterations-in-PBMCs-associated-with-T2DM.

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

## Acknowledgements
This research was supported by funding from ETH Zürich and PSI to GVS. We also thank Celestino Padeste for useful discussions related to device fabrication. CU acknowledges support from NCCIH/NIH (1DP2AT012345) and ONR (N00014-22-1-2116). Parts of Figs. 1 and 5 were created using biorender.com tools. We would also like to thank Dr. Elvira Forte and Dr. Anne Shim for editing our manuscript.

## Author contributions
M.M.A., R.G., I.F., C.U., and G.V.S. designed the research. I.F. performed the cohort selection and sample collection. M.M.A. designed and fabricated microfluidic devices. M.M.A., R.G., and G.V.S. developed the protocol for live PBMC imaging and compression described in the paper. M.M.A. implemented the protocol for live PBMC purification and collected light microscopy data for the clinical samples. R.G., C.U., and G.V.S. developed the computational pipeline to analyze the chromatin imaging data. R.G. implemented and applied the computational analysis described in the paper for the clinical samples. R.G., M.M.A., and G.V.S. wrote the manuscript. M.M.A., R.G., I.F., C.U., and G.V.S. have read and reviewed the manuscript.

## Funding

## Competing interests
The authors declare no competing interests.
