## [Transparent Peer Review file · Communications Medicine]

Detecting chromatin state alterations in PBMCs associated with Type 2 Diabetes Mellitus

Corresponding Author: Professor G.V. Shivashankar

Version 0:

Reviewer comments:

Reviewer #1

(Remarks to the Author)

This study by Afarani et al demonstrates chromatin imaging of blood cells in different stages of diabetes. This is an interesting concept with an important aim of understanding patient-specific immune dysfunction in T2D. Bioinformatic approach of analysing chromatin images is unique. However, the writing is complex and lacks explanation/refers to other studies from the group, making interpretations of results difficult. I strongly recommend expanding results and discussion with special focus on simplifying the message. The title and abstract indicate altered immune subpopulations, which is misleading as authors have identified 9 clusters, however there is no information on which immune cells are present in these clusters. Having this information would have been much more meaningful. The importance of having such imaging approach on a POC technology is also unclear, especially if it is unknown as to which immune cells are changing in T2D. Having this knowledge would have led to implementing immune-targeting therapies specifically to those people with altered immune subsets. I therefore fail to understand the significance of this methodology in T2D care and management. Additional comments are below:

Major:

- 1) POC: Ideally, a POC is a device that provides rapid output using easily operated (even by patients) assay. I do not think the current method of chromatin imaging is a POC. It needs PBMC isolation using specific laboratory methods as well as use of confocal microscopy to see the changes. I recommend that authors should not mention this as a POC assay. The method and its potential use are scientifically interesting; however, it is not a POC assay.
- 2) Introduction: suggests the need to address "patient-to-patient variability in T2D immune dysfunction". It should also elaborate either in intro or in discussion on why this is important. Discussion can also add on current methods of immune function assessment and their limitations, with appropriate references.
- 3) Fig 1 or Table 1: Provide T2D duration. A correlation of sex, age and duration of T2D with the imaging would be insightful.
- 4) Is the staining in Figure S1B-S5A demonstrate Hoechst and DRAQ7 stains (left panels)? Add this information to the legend. The segmentation colouring (right panels) has different colours. What do they represent? Is it just for visual effect or do they present some scaling effect based on Hoechst dye intensity? If they do not present anything relevant for data analysis, I suggest removing them as they do not add any additional information but might be confusing for some readers.
- 5) Can authors provide any information to understand the identity of immune cell types within the 9 clusters? Specifically, cluster 1 and 6 that is different between T2D and healthy? I think this is a major concern.
- 6) Panel 2G needs more elaboration in results section. What is MeD vs PBMC subpopulation? Explain what is meant by "disease-specific enrichment" and "cluster-specific enrichment"? Don't just describe the method of obtaining those plots but what does that indicate and how those two enrichments are different.
- 7) Handcrafted features: which features were the most important in separating the disease stages? Add some details of these in the results section.
- 8) Fig S9: check the legend for accuracy. What is the difference between panels A, B and C? Are they different features used in each heatmap? Why are there only "control" individuals mapped in this figure?
- 9) Fig S10: why are there 15 clusters here? What features are different between Fig S9 and Fig S10 heatmaps?
- 10) Authors have selected CD25 as PBMC activation marker. Although CD25 is a known marker for activation, have you tested other activation markers such as CD69, HLA-DR to confirm your findings?
- 11) Fig S12C and Fig 5: how higher heterochromatin/H3K9Me3/gene silencing is associated with more activation? What could be the mechanism? Is it that CD25 is probably indicating a specific subset of immune cells rather than presenting an overall activation?
- 12) Fig 5: I wonder why the data for Lamin, H3K9Me3 and CD25 in prediabetes not presented?

- 13) Methods: Should include criteria of defining prediabetes and T2D.
- 14) Methods: use of rabbit anti- γ -H2AX is mentioned for study of DNA damage. However, I do not find any data for this.
- 15) Discussion should include thoughts on the top important features identified to be segregating disease states and how they might be linked/changed with disease states.
- 16) Authors need to discuss study strengths, but more so the limitations including validation of their algorithms in an independent cohort, associations with other co-morbidities, diabetes complications (if known), lack of immune cell analyses, functional analysis of the immune cells, etc.

Minor:

- 17) Title and abstract should reflect the actual message. Please modify.
- 18) "Tracking inflammation during the T2DM progression with sequencing and proteomics techniques remains costly, labor-intensive, and impractical". Provide a reference.
- 19) Intro: "In summary, we demonstrate an affordable, compact, scalable, point-of-care (POC) chromatin imaging assay to monitor immune dysfunction in T2DM management for personalized disease tracking." I think this is an overstatement. Change and reflect what exactly is shown in this paper.
- 20) "Although chromatin images from individual T2DM stages looked cluster 6 showed enrichment in diabetic individuals (Fig. S6A, S6B)". The reference of figures is incorrect here.
- 21) Discussion: Ref#25 is very specific to a particular region and the statement "Inflammatory biomarkers such as C-reactive protein (CRP), (IL6) show significant heterogeneity across patients [25]" would need more global evidence.

Reviewer #2

(Remarks to the Author)

This study presents a novel imaging-based assay to detect immune alterations across Type 2 Diabetes Mellitus (T2DM) stages by profiling chromatin architecture in live PBMCs. By integrating 3D chromatin imaging with a variational autoencoder and clustering, the authors identify nine chromatin states differentially enriched across healthy, prediabetic, and diabetic individuals. These chromatin features, along with nuclear mechanical properties and marker expression, support the potential of chromatin structure as a functional immune biomarker. While the methodology is technically innovative and the assay is proposed as scalable, its clinical applicability is limited by small sample size, lack of immune cell-type validation, and infrastructural complexity. A revision to address these issues will improve this work before publication.

Major comments:

1. While the model demonstrates promising classification accuracy, it currently functions as a black-box predictor without interpretable or actionable thresholds. For clinical relevance, the authors should consider providing calibrated probability scores, ROC curves, or defining rule-based chromatin signatures (e.g., percent of cells in Cluster 6) that can be used to guide diagnostic decisions. Can the authors clarify whether such thresholds were explored or could be derived from their enrichment features?
2. Was a power analysis conducted to justify the sample size ($n=57$)? While downsampling and cross-validation are applied, the authors do not statistically justify that the dataset is powered to detect meaningful inter-group differences.
3. The random forest classifier achieves AUCs $>80\%$ and even 100% in some comparisons, raising concerns about overfitting in a small dataset. The manuscript would benefit from additional validation metrics (e.g., permutation tests, independent test set, confidence interval) to assess robustness and generalizability to larger or more diverse populations.
4. Chromatin condensation or nuclear changes are not specific to T2DM, they can also occur in response to aging, infection, cancer, or stress. The study does not establish mechanistic links between observed chromatin states and T2DM pathophysiology (e.g., beta-cell function, insulin resistance, or metabolic inflammation). The authors acknowledge that nuclear changes may reflect immune activation but don't tie them mechanistically to diabetes-specific processes.
5. The identification of nine chromatin clusters is interesting, particularly the enrichment of Clusters 1 and 6 in specific disease stages. However, the biological significance of these clusters remains unclear. Are they associated with known PBMC subtypes (e.g., T cells, monocytes)? Flow cytometry or transcriptomic validation would considerably strengthen this link.
6. How does the proposed chromatin imaging assay compare to established biomarkers like HbA1c, CRP, or IL-6 in terms of diagnostic power, cost, ease of use, and scalability? Can the authors discuss its integration into current clinical workflows or its potential complementarity?

Minor comments:

1. The term "point-of-care" is used throughout the manuscript. However, the workflow still involves confocal imaging and machine learning, which may not yet be feasible in decentralized clinical settings. Clarify whether this term refers to future potential or current readiness.
2. The authors mention future release of code and data. Please specify a timeline or repository (e.g., GitHub) to ensure reproducibility.

Reviewer #3

(Remarks to the Author)

Dr. Afarani et al. present a novel assay using chromatin imaging to monitor immune dysfunction in T2D. They studied 57 individuals across healthy, prediabetic, and diabetic stages, using microfluidic imaging of live PBMCs. Through representation learning on chromatin images, they identified nine PBMC clusters, with some PBMC subsets predictive of T2DM. Diabetic PBMCs exhibited altered nuclear and chromatin mechanics, decreased Lamin A/C expression, and increased cellular activation. The approach is interesting and potentially of use in different contexts of disease monitoring. However, I have a number of concerns:

Introduction:

- The authors state that tissue alterations in the adipose tissue microenvironment send “chemical signals” into the bloodstream, influencing the function of other organs. This concept requires elaboration and mechanistic detail. Please expand on how adipose tissue-derived factors (e.g., adipokines, cytokines, exosomes) influence systemic immune alterations (doi: 10.1016/j.isci.2024.110528).
- Furthermore, partial exhaustion of tissue-infiltrating T cells has been demonstrated in T2DM and should be discussed (e.g., doi: 10.1172/jci.insight.139793; doi: 10.1016/j.isci.2024.109032). Additionally, resistance of T cells to cytokine-mediated suppression is relevant (doi: 10.1136/bmjdr-2019-000772).
- The authors mention: “Recent multi-omics studies have identified significant shifts in leukocyte subpopulations and gene expression patterns throughout the progression of T2DM [8,9].” Please provide appropriate references to support this statement. It would be interesting to mention multi-omics based approach in other diabetes subtypes (DOI: 10.1038/s43856-025-00922-7)
- The sentence: “Our findings indicate that specific PBMC subpopulations reveal patient-specific alterations in immune cells as T2DM advances.” is imprecise because there is no longitudinal follow-up of the same patients. Please rephrase this statement to reflect that differences were observed across groups, rather than progression within individuals.

Results:

-Diabetic and prediabetic groups show clearer separation (e.g., Fig. 2G), while discrimination of healthy individuals is less accurate. How do you explain this, and how might it be improved?

-How do you explain the observed changes in nuclear size and mechanical properties of PBMCs in T2DM? Have you controlled for differences in cell subset composition? For example, increased monocytes, dendritic cells, or residual neutrophils could contribute to these observations (besides the cell activation that the authors show).

-In Fig. 5 and S12 the authors conclude: “Together, these observations provide a functional basis for the use of chromatin architecture as a biomarker to track T2DM-related immune alterations.” However, the correlation of nuclear areas with Lamin A/C, H3K9Me3, and CD25 expression does not in itself establish a functional role. Please clarify this point and avoid overinterpretation.

-The authors describe the evidence as “alterations during T2DM progression” throughout the manuscript. Since this is a cross-sectional study, no conclusions about disease progression can be drawn. Please revise the language to avoid implying longitudinal data.

-The authors conclude that “specific immune cell subpopulations (Clusters 1 and 6) are enriched in patients with T2DM and healthy individuals.” Could the authors describe these populations phenotypically (e.g., by flow cytometry markers or transcriptomic signatures)? Knowing which clusters are enriched is less informative without understanding their phenotypic and functional properties.

Discussion:

-The authors state that the proposed live chromatin imaging provides a clinically relevant readout of immune alterations. However, no detailed cell subset characterization is possible with this technology. What is the clinical relevance of detecting these cell clusters without phenotypic information, especially in the context of POC technologies? Furthermore, the authors mention that inflammatory biomarkers show significant heterogeneity across patients; however, no data on the heterogeneity of the assay results are provided. Also, the cohort size is relatively small; please discuss this limitation.

-The authors mention the cost-effectiveness of their technology. It would be useful to compare costs with established biomarker assays to better contextualize this claim.

Version 1:

Reviewer comments:

Reviewer #1

(Remarks to the Author)

Reviewer #3

(Remarks to the Author)

the authors have extensively replied to all my comments and concerned.

Response to Reviewer Comments for COMMSMED-25-0700-T: Detecting immune alterations in Type 2 Diabetes Mellitus patients using PBMC chromatin imaging biomarkers

Maryam Moazeni Afarani, Rajshikhar Gupta, Caroline Uhler,
Issa Fetian, GV Shivashankar

We thank the editor and all the reviewers for their consideration and comments/questions regarding our manuscript. All the Reviewers have raised several important, but addressable questions, and made helpful suggestions to improve the manuscript for clarity, engagement, and impact.

In what follows, we have reproduced the referee comments (in full) in black text, our responses in **Author reply: blue** text, and changes to the manuscript are quoted in **Revisions: teal** text. The location where text has been changed or added is mentioned in terms of the relevant section, as well as page numbers and location as in the revised, marked manuscript.

Reviewer 1 (Remarks to the Author):

This study by Afarani et al demonstrates chromatin imaging of blood cells in different stages of diabetes. This is an interesting concept with an important aim of understanding patient-specific immune dysfunction in T2D. Bioinformatic approach of analysing chromatin images is unique. However, the writing is complex and lacks explanation/refers to other studies from the group, making interpretations of results difficult. I strongly recommend expanding results and discussion with special focus on simplifying the message. The title and abstract indicate altered immune subpopulations, which is misleading as authors have identified 9 clusters, however there is no information on which immune cells are present in these clusters. Having this information would have been much more meaningful. The importance of having such imaging approach on a POC technology is also unclear, especially if it is unknown as to which immune cells are changing in T2D. Having this knowledge would have led to implementing immune-targeting therapies specifically to those people with altered immune subsets. I therefore fail to understand the significance of this methodology in T2D care and management. Additional comments are below:

Author reply: We thank the reviewer for their comments and feedback regarding our manuscript. In response, we have revised the title and abstract of the manuscript to better align with the findings of our study. We have added a section in the Discussion to compare our technique with established methods. We revised the language of the Results section to simplify the message for a broad audience. Traditional methods for tracking immune alterations in T2DM, such as C-reactive protein (CRP) and IL-6 screening, remain limited in sensitivity and specificity, even within controlled cohorts [37, 19, 36]. Moreover, even when combined into multi-biomarker

panels, these established approaches provide only modest accuracy in identifying at-risk populations [29]. In this study, we demonstrate an orthogonal imaging-based assay that measures alterations in chromatin states and shows promising results in a cohort of 57 individuals. The reviewer’s point about the importance of identifying specific immune-cell subsets is well-taken. We have revised the Discussion to emphasize that future studies integrating chromatin imaging with single-cell sequencing or proteomics will be critical to extend our findings and enhance the clinical utility of this methodology in T2D care and management.

Major Comments:

1. POC: Ideally, a POC is a device that provides rapid output using easily operated (even by patients) assay. I do not think the current method of chromatin imaging is a POC. It needs PBMC isolation using specific laboratory methods as well as use of confocal microscopy to see the changes. I recommend that authors should not mention this as a POC assay. The method and its potential use are scientifically interesting; however, it is not a POC assay.
-

Author reply: We agree with the reviewer’s comment. Our current workflow does rely on research infrastructure, including confocal imaging, and is therefore not directly translatable as a point-of-care device at this stage. We acknowledge that in future work, we will adapt each of these components for true point-of-care applicability. To avoid overstating the current readiness, we have removed the term ‘point-of-care’ from the manuscript.

Revisions: We have made the following changes to the previous version of the manuscript to reflect this:

Previous version at line 6: Type 2 Diabetes Mellitus (T2DM) involves patient-specific immune dysfunction not addressed by current point-of-care (POC) technologies.

Present version at line 6: Type 2 Diabetes Mellitus (T2DM) involves patient-specific immune dysfunction not addressed by current diabetes management devices.

Previous version at line 8: Here we present a chromatin imaging-based POC assay to monitor immune dysfunction during T2DM progression.

Present version at line 8: Here we present a chromatin imaging-based assay to monitor PBMC subpopulation stratification during T2DM progression

Previous version at line 21: While recent advances in point-of-care (POC) technology have enhanced the accuracy and miniaturization of diabetes management tools such as insulin delivery systems and blood glucose monitors [2, 9], current POC devices do not address patient-to-patient variability in Type 2 Diabetes Mellitus (T2DM) related immune dysfunction.

Present version at line 20: While recent advances in diabetes care and management technology have enhanced miniaturization and accessibility, such as insulin delivery systems and blood glucose monitors [2, 9], they do not address patient-to-patient variability in Type 2 Diabetes Mellitus (T2DM) related immune dysfunction.

Previous version at line 70: In summary, we demonstrate an affordable, compact, scalable, point-of-care (POC) chromatin imaging assay to monitor immune dysfunction in T2DM

management for personalized disease tracking.

Present version at line 72: In summary, we demonstrate that chromatin imaging assay can be an affordable, compact, and scalable approach for potential clinical use in monitoring PBMC subpopulation stratification relevant to T2DM management and personalized disease tracking.

2. Introduction: suggests the need to address “patient-to-patient variability in T2D immune dysfunction”. It should also elaborate either in intro or in discussion on why this is important. Discussion can also add on current methods of immune function assessment and their limitations, with appropriate references.
-

Author reply: We agree with the reviewer’s comment and have revised the Introduction to more explicitly highlight examples of specific complications arising from immune responses in patients with certain comorbidities. This comment is particularly relevant to our study cohort, where many patients present with multiple comorbidities, as illustrated in Fig. 1C. Furthermore, we have expanded the Discussion section to include a comparison of our method with commonly used clinical approaches such as HbA1c, CRP, and IL6 profiling for immune function assessment and their limitations, with appropriate references.

Revisions: **We have modified the introduction to reflect the above comment on line 32 of the revised manuscript:**

Together with infiltrating subpopulations of NK and B lymphocytes [17, 51], these M1 macrophages amplify inflammation in the microenvironment of VAT and other organs, leading to patient-specific comorbidities [35, 7].

In the liver, adipose-derived cytokines and immune cell trafficking activate Kupffer cells and recruit monocytes and T cells, promoting NASH and fibrosis in specific patient cohorts [49]. Similarly, in certain patient subpopulations, elevated levels of VAT-derived mediators (e.g., MCP-1, TNF- α , IL-6, IL-18) drive macrophage and T-cell infiltration into glomeruli and tubules, fueling chronic inflammation and fibrosis that characterize diabetic nephropathy [50]. Inclusively, in pancreatic islets, macrophage-derived IL-1 β and TNF- α impair insulin secretion and reduce β -cell survival, linking adipose inflammation to endocrine dysfunction in T2DM [18].

We have modified the discussion section to reflect the above comment on line 308 of the revised manuscript:

When applied to our cohort, our approach has the specificity of 0.78 (CI: 0.4 - 0.972) and sensitivity of 0.75 (CI: 0.194 - 0.994) in distinguishing diabetic individuals. In this context, HbA1c with well-validated cut-offs (HbA1c $\geq 6.5\%$) show high specificity ($\geq 90\%$), but moderate sensitivity (45-70%) [19, 26, 3]. CRP and IL-6 are established inflammatory markers associated with increased risk of T2DM (e.g., CRP relative risk 4.2, IL-6 relative risk 2.3) [37], and in multivariate models combined with other biomarkers (for eg. adiponectin, ferritin, interleukin-2 receptor A (IL2RA), glucose, and insulin) achieve accuracies around 76–78% in predicting at risk patients [29] (Table S3). From a cost perspective, HbA1c and CRP are low-cost assays [15, 22], while IL-6 is substantially more expensive [8] (Table S3). Our approach remains inexpensive and uses low-cost reagents such as Hoechst dye and microfluidics devices [21], suggesting material costs could be comparable with scalability (Table S3). Importantly, our approach provides complementary

information about the functional state of the immune cell that is not directly addressed by established biomarkers.

We have summarized these points in Table S3:

Dimension	HbA1c	CRP / IL-6	Chromatin imaging (this work)
Primary signal	Long-term glycemia (2-3 months) [3], non-indicative of patient-specific inflammation	Systemic inflammation [29]	Alteration in chromatin organization in PBMCs due to sub-population stratification, activation or senescence
Clinical role	Diagnostic & monitoring standard	Adjunct risk / inflammation marker	Exploratory/adjunct; Orthogonal to established methods
Diagnostic power	Established cut-off: HbA _{1c} ≥ 6.5%; high specificity (≥ 90% [19]); Moderate sensitivity (45.5%[19], 70%[26])	Associated with risk but non-specific [37, 29] CRP: 1.5–12.0 (rel. risk = 4.2); IL-6 Range: 0.9–5.6 (rel. risk = 2.3) [37], multivariate accuracy 76–78% [29]	specificity of 0.78 (CI: 0.4 - 0.972) and sensitivity of 0.75 (CI: 0.194 - 0.994) in distinguishing diabetic individuals.
Cost per test (relative)	Low (\$8 – \$16) [15]	CRP: Low (\$12 – \$16)[22]; IL-6: Moderate: (\$100 - \$300) [8]	Low (Estimated with Scalability): ≤ \$10; includes low cost reagents like Hoechst (\$143/10 mL Thermo Fischer) & Microfluidics Devices [21]
Capital/equipment	Standard clinical analyzer	Standard immunoassay platforms	Benchttop centrifuge + density-gradient separation + confocal microscope (or high-NA wide-field) + compute for ML
Ease of use	High	High	automation feasible
Throughput / TAT	High / same-day	High / hours	Moderate (same-day feasible); automation can increase throughput
Scalability	Mature	Mature	Scalable with plate-based imaging, automated segmentation, model deployment; not POC yet
Complementarity	Glycemic exposure	Inflammatory burden	Promising sensitive and specific approach in the context of this cohort size; orthogonal to established methods.

Table 1: Comparison of HbA1c, CRP/IL-6, and chromatin imaging assay across diagnostic and practical dimensions.

-
3. Fig 1 or Table 1: Provide T2D duration. A correlation of sex, age and duration of T2D with the imaging would be insightful.
-

Author reply: We appreciate the reviewer’s suggestion to examine the effects of T2D duration. Duration of T2D (time since diagnosis) was not provided for most participants, precluding a reliable analysis of duration–imaging associations. Using available covariates, we tested associations with clusters enrichment score of Clusters C_{HFE} 4, 5, 13, and 11 with age and sex across diagnostic groups. The results (Fig. R1 shown below, which we added to the revised manuscript as Fig. S16), show no associations with age. However, in clusters enrichment score of prediabetic and diabetic cohorts, we do see some differences due to sex in the median values as shown in Fig. R2 below, which we added as Fig. S17 to the revised manuscript. Although the association was not significant after correction for false discovery rate

Revisions: We added the following statement in the Discussion section on line 293 of the revised manuscript:

Figure R1: (corresponding to the new Fig. S16 in the revised manuscript) A-Q Scatter plot annotated with Pearson R and corresponding p-value with Age, color coded for individuals with different pathologies.

Using the available covariates (age, sex), we tested associations between cluster–enrichment scores (derived from handcrafted features) and these covariates within and across diagnostic groups. We observed no meaningful correlation with age (Pearson’s $R \leq 0.182$; Fig. S16A–Q). In addition, while there are some sex-related differences in the cluster enrichment scores of prediabetic and diabetic cohorts, these differences are not statistically significant after controlling for false discovery rate (FDR) using Benjamini-Hochberg correction (Fig. S17A–C).

Figure R2: (corresponding to the new Fig. S17 in the revised manuscript) A-C Box plot showing association with Sex with individuals with different pathologies. p Value (Welch) unless stated is non-significant, p Adj (Benjamini-Hochberg).

-
4. Is the staining in Figure S1B–S5A demonstrate Hoechst and DRAQ7 stains (left panels)? Add this information to the legend. The segmentation colouring (right panels) has different colours. What do they represent? Is it just for visual effect or do they present some scaling effect based on Hoechst dye intensity? If they do not present anything relevant for data analysis, I suggest removing them as they do not add any additional information but might be confusing for some readers.
-

Author reply: We agree that the Figures S1B–S5A lacked clarity. These figures present side-by-side representative examples of data from each individual included in the study: the left panel shows 2D projections of 3D confocal images of Hoechst-stained live PBMC nuclei, while the right panel displays the corresponding nuclear segmentation. Since our analysis relies on nuclear morphological features, we wanted to demonstrate that the segmentation algorithm—described in the Methods section and illustrated in Figure S1A—performs well across live chromatin imaging data from different individuals. We also wanted to provide readers with a visual overview of the dataset generated. We have clarified this purpose more explicitly in the revised manuscript.

Revisions: **We have update the figure captions for Figure S1B - S5A:**

Previous version: Representative maximum z projected confocal images (left) and corresponding segmentations (right). The segmentation is colormapped (colormap: jet) and randomly shuffled to enhance visual distinction and clarity. The number annotations to images represent corresponding Sample No (see Table 1.)

Revised version: Representative maximum z projected confocal images stained with

Hoechst (left) and corresponding nuclear segmentations (right). The segmentation is colormapped (colormap: jet) to enhance visual distinction and clarity. The number annotations to images represent the corresponding Sample No (see Table 1.)

5. Can authors provide any information to understand the identity of immune cell types within the 9 clusters? Specifically, cluster 1 and 6 that is different between T2D and healthy? I think this is a major concern.
-

Author reply: We would like to clarify that the primary aim of the present study is to show that chromatin imaging of peripheral blood cells is sensitive and specific to T2D-associated alterations, and that these image-derived features can *complement* established immune dysregulation biomarkers (e.g., IL-6, CRP). Because our cohort does not include matched immunophenotyping or transcriptomic measurements, we cannot ascribe the nine chromatin clusters to canonical PBMC lineages (e.g., T cells, monocytes) with confidence. We view these experiments as a natural next step but beyond the scope of the current manuscript. We have clarified this limitation in the revised text.

Revisions: **We have added the following text on line 326 of the revised manuscript:** Nevertheless, several limitations should be noted. For instance, our algorithms were developed and tested in a single cohort, and independent validation in larger, multi-center cohorts will be required to establish generalizability. The dependence on research lab infrastructure (e.g. centrifuge, confocal microscope, workflow complexity), can be addressed by developing specialized automated imaging and screening workflows. Moreover, our approach does not provide detailed immune cell subpopulation resolution, nor did we perform functional assays (e.g., cytokine secretion, T cell or macrophage activation studies) to mechanistically link chromatin architecture to immune dysfunction. Future studies integrating chromatin imaging with single-cell sequencing or proteomics will be critical to further validate and extend our findings.

6. Panel 2G needs more elaboration in results section. What is MeD vs PBMC subpopulation? Explain what is meant by “disease-specific enrichment” and “cluster-specific enrichment”? Don’t just describe the method of obtaining those plots but what does that indicate and how those two enrichments are different.
-

Author reply: We agree with the reviewer that our description in the caption and panel subheadings of Fig. 2G were ambiguous and inconsistent. We have elaborated on the description of Figure 2G in the figure caption and the Results section. We have also removed the term “MeD vs PBMC subpopulation” from everywhere in the manuscript.

Revisions:

Fig 2. PBMC chromatin states at different stages of T2DM.

Previous version

(G) (left) Disease stage-specific enrichment profile, $Enrichment_d$ (see Methods), across

different clusters and (right) Cluster-specific enrichment profile, $Enrichment_h$ (see Methods), across different stages of T2DM, respectively.

Revised Version

(G) (left) Cluster-specific enrichment profile, $Enrichment_h$ (see Methods), representing the fraction of cells from each T2DM disease stage in a cluster. Warmer colors represent a higher fraction of a particular pathology in a cluster. Row sums to 1. (right) Disease-specific enrichment profile, $Enrichment_d$ (see Methods), representing fraction of cells from different clusters in a disease stage (colder colors across all columns (for eg, 9 and 2) representing less populous and smaller clusters). Columns sum to 1.

-
7. Handcrafted features: which features were the most important in separating the disease stages? Add some details of these in the results section.
-

Author reply: As per the reviewer’s comment, we have revised the manuscript to add the details regarding the analysis of chromatin states using our interpretable handcrafted features. This analysis is shown in Figure R3 below, which we added as Fig. S9 to the revised manuscript. In particular, we evaluated the most important features for pathology class discrimination by GINI importance score, as shown in Figure R3B. We find that the heterochromatin organization features describing the radial location of heterochromatin domains from the center “D3” and separation of heterochromatin domains from the nuclear boundary represented by annotation “D4” are found to be strongly associated with pathology class identification (Figure R3C). We have also highlighted these features in Figure 2D-E in the main manuscript. Expounding on this further, we have aimed to provide interpretability to VAE-embedding-based clusters C_{VAE} (Figure R3K). Based on the reviewer’s comment, we have also added the details of cluster enrichment for C_{HFE} clusters identified through handcrafted embedding in the Results section, given their direct interpretability. We thank the reviewer for this insightful comment.

Revisions: **We have added the following sentences on line 132 of the revised manuscript.**

We evaluated the most important features for disease stage discrimination by GINI importance score, as shown in the Fig. S9B. Among the handcrafted descriptors, we find that the heterochromatin organization features describing the radial location of heterochromatin domains from the center and separation of heterochromatin domains from the nuclear boundary represented by annotation are found to be strongly associated with pathology class (Fig. S9C). The embedded features obtained from the VAE also capture these key chromatin organization features, as shown in Fig. 2D-E.

We have also discussed the aspect of the Handcrafted feature on line 192 of the revised manuscript.

Notably, C_{HFE} 5 enriches for nuclei with maximum radial location of heterochromatin domains from the center “D3” and maximum separation of heterochromatin regions from the nuclear boundary represented by annotation “D4”, shown in Fig S9 J.

Figure R3: (corresponding to the new Fig. S9 in the revised manuscript) (A) Mean recall (\pm s.e.m.) across 5-fold cross-validation for each pathology class (Healthy, Prediabetes, Diabetes), shown alongside a permutation-null baseline obtained by shuffling pathology labels before each random train/test split. (B) Random-forest Gini (mean decrease in impurity) feature importances for pathology classification, averaged over random splits. (C) Top handcrafted features capturing heterochromatin organization of nuclei (e.g., texture, intensity, radial distribution), ranked by importance. (D) Sizes of clusters obtained from iterative dense-consensus Leiden clustering in the VAE embedding C_{VAE} ; mean Adjusted Rand Index across runs = 0.85 ± 0.102 . (E) Cluster-level consensus scores after iteratively removing nuclei from clusters with size < 30 . (F) Cross-method association matrix showing, for each VAE-derived cluster labels, C_{VAE} (rows), the fraction of nuclei assigned to each handcrafted feature cluster label, C_{HFE} (columns); rows sum to 1. (G) Sizes of clusters obtained from iterative dense-consensus Leiden clustering in the Hand-crafted feature embedding C_{HFE} ; mean Adjusted Rand Index across runs = 0.91 ± 0.016 . (H) Cluster-level consensus scores in C_{HFE} after iteratively removing nuclei from clusters with size < 20 . (I) (left) Cluster-specific enrichment profile, $Enrichment_h$ (see Methods), representing the fraction of cells from each T2DM disease stage (column) in a specific cluster (rows). Warmer colors represent a higher fraction of a particular disease stage in a cluster. Row sums to 1. (right) Disease-specific enrichment profile, $Enrichment_d$ (see Methods), representing fraction of cells from different clusters in a disease stage. Colder colors across all columns (for eg, 14 and 15) represent less populous and smaller clusters. Columns sum to 1. (columns), the fraction of nuclei assigned to each VAE cluster label, C_{VAE} (rows); columns sum to 1. Differential features among clusters identified from handcrafted features (J) and VAE-derived clusters (K).

8. Fig S9: check the legend for accuracy. What is the difference between panels A, B and C? Are they different features used in each heatmap? Why are there only “control” individuals mapped in this figure?

Author reply: In Fig. S9 of the previous manuscript (shown below as Figure R4 for ease of reference), each identifier (e.g., “65-control”) refers to the same individual across panels. The intensity in the rows in the heatmaps in Fig. R4A–C represents the fraction of total nuclei in each cluster for a particular individual. Since the cohort population did not change between different comparisons, the fraction of nuclei in each cluster for a particular individual has not changed in the heatmaps A, B, and C. The apparent differences arise only from a change in the order of cluster labels shown as column headers. The “control” designation indicates that these nuclei were imaged without compression. Parallel experiments were performed under compression for each dataset; however, we did not observe any significant differences in enrichment, and therefore, those results are not presented. The first row header denotes the actual pathology of the individual, and the second row header denotes the predicted pathology of the individual. These predictions changed in each comparison depending on whether we had a 3-way classification between healthy, prediabetic, and diabetic individuals, or two-way classifications between healthy vs diabetic R4A or prediabetic vs diabetic R4B. We provided these heatmaps to visualize

Figure R4: (corresponding to the old Fig. S9) VAE obtained features and embedding: (A) & (B) Heatmap where each row indicates enrichment vector of nuclei from an individual across different cluster labels. Each row head annotations indicate the (left) ground truth and (right) predicted disease stage of T2DM. The color map indicates the fraction of nuclei from PBMCs of individual samples in a particular cluster. (C) The 3×3 confusion matrix indicates the 3-way classification of individual samples from diabetes, prediabetes, and healthy individuals upon leaving one out cross-validation for RFC trained on the enrichment profile of remaining samples. (D) The 2×2 confusion matrix indicates the 2-way classification of individual samples from prediabetes, and healthy individuals upon leaving one out cross-validation for RFC trained on the enrichment profile $Enrichment_i$ of remaining samples.

which cluster is responsible for true-positive predictions. We thank the reviewer for pointing out the editing error in our captions for Fig. S9 of the previous version of the

manuscript. We have also cleaned up our labels in the revised manuscript to avoid any confusion and removed the "control" word from the figure labels.

Revisions: We have expanded the analysis presented in the previous Fig. S9 and renamed the updated Figure S10, which is shown in Fig. R5 below. We have also revised the figure legend accordingly.

Figure R5: (corresponding to the updated Fig. S10 in the revised manuscript) VAE obtained features and embedding: Heatmap where each row indicates enrichment vector of nuclei from individuals from (A) Healthy and Diabetic, (B) Diabetic and Prediabetic, (C) Healthy and Prediabetic, and (D) All three pathologies across different cluster labels. Each row head annotations indicate the (left) ground truth and (right) predicted pathology. The color map indicates the fraction of nuclei from PBMCs of individual samples in a particular cluster. (E-F) The bar plots indicate the TPR for binary Pathology label predictions averaged over 5-fold cross-validation against randomly shuffled samples. (G) The 2×2 confusion matrix indicates the TPR for LOOCV classification of healthy vs prediabetic individuals. The ROC curve indicates the probability of TP vs FP for 5-fold cross-validation and prediction of individual pathology if supplied with (H) Diabetes vs prediabetes vs healthy (I) Diabetes vs prediabetes enrichment vectors. (J) Bar plot indicates the recall for class label predictions averaged over 5-fold cross-validation against randomly shuffled samples. (K-L) Box plot representing Enrichment score calculated for Cluster 1 & 6 across different pathologies. Each dot represents an individual.

9. Fig S10: why are there 15 clusters here? What features are different between Fig. S9 and Fig. S10 heatmaps?

Author reply: We thank the reviewer for this comment. Fig. S9 and S10 in the previous version of the manuscript were based on two distinct feature sets: Fig. S9 shows clustering derived from VAE-based latent features [55], whereas Fig. S10 is based on handcrafted chromatin features [46, 14]. The plausible differences in the number of distinct chromatin clusters could be because the VAE embedding is obtained from the 2D segmentation of max z projected images, whereas the handcrafted features were calculated based on 3D nuclear volume segmentations. To strictly determine the number of clusters for each embedding and robustly assign cluster labels to each nucleus, we have extended our method of clustering the nuclei. We have employed consensus clustering [31] and evaluated the cluster assignment using the adjusted Rand index. We iteratively removed the nuclei that lie in a cluster with a size value less than 30 until the cluster length and size converged, or if we achieved mean ARI ≥ 0.8 . These nuclei that were not localized in any distinct C_{VAE} cluster were assigned the 'UC' label; they accounted for 7.07% of the total population. Given this additional validation step, we have reevaluated the previously made comparisons among different cohorts of the study and updated the individual figure panels at their respective places. The evaluated ARI score and cluster sizes are shown in Fig. R3D,E for the VAE embedding and Fig. R3G,H for the handcrafted feature embedding in the revised manuscript. We find that there is some association in cluster identities assigned by these two parallel methodologies, indicated by the small fraction of nuclei sharing the same cluster labels in C_{VAE} and C_{HFE} , as shown in Fig. S9I. Moreover, individual clusters derived from VAE (C_{VAE}) or handcrafted (C_{HFE}) embeddings enrich nuclei with different morphometrical and chromatin organization characteristics, as shown in Fig. S9J,K.

Revisions:

We have added the following sentences on line 140 of the revised manuscript.

Notably, we find that C_{VAE} clusters 6, 7, and 8 are particularly enriched with PBMC chromatin images from diabetic samples, cluster 1 is enriched with nuclei from prediabetic samples, and cluster 4 is enriched with nuclei from healthy samples (Fig. 2G).

The identified clusters were robust to variations in initial conditions, achieving an adjusted Rand index (ARI) of 0.85 ± 0.10 (Fig. S9D) after excluding nuclei that did not cluster. These nuclei, localized outside of the 9 distinct C_{VAE} clusters, henceforth labelled as "UC", accounted for 7.07% of total cells. Motivated by assigning interpretable quantitative descriptors of alterations to chromatin state, we reevaluated cluster labels of nuclei using handcrafted features. We derive 15 distinct C_{HFE} clusters as shown in Fig. S9 F, with C_{HFE} clusters 4, 11, 12, 13, 15 and C_{HFE} clusters 0, 2, 5, 6, 8 consisting of a higher fraction of nuclei belonging to prediabetic and diabetic individuals. The C_{HFE} cluster remarkably high adjusted Rand index of 0.91 ± 0.01 , agnostic to initial conditions (Fig. S9 G, H). We find that there is some association in cluster identities assigned by these two parallel methodologies, indicated by the small fraction of nuclei sharing the same cluster labels in C_{VAE} and C_{HFE} , as shown in Fig. S9I. Moreover, individual clusters derived from VAE (C_{VAE}) or Handcrafted (C_{HFE}) embeddings enrich nuclei with different morphometrical and chromatin organization characteristics, as shown in Fig S9 J & K. **Specific updates to Fig. 2, Fig. 3, Fig. S10 (which was Fig. S9 in the previous version), and Fig. S11 (which was Fig. S10 in the previous version) are described below.**

- (a) Fig 2A, reproduced here as Fig. R6A. The UMAP is updated with new cluster assignment labels as shown in Fig. R7A.
- (b) Fig 2B, reproduced here as Fig. R6B. The representative images from each cluster

Figure R6: (corresponding to the old Fig. 2)

Figure R7: (corresponding to the updated Fig. 2 in the revised manuscript)

Figure R8: (corresponding to the old Fig. 3)

were updated with new cluster IDs as shown in Fig. R7B.

- (c) Fig. 2G, reproduced here as Fig. R6G, The heatmaps showing Cluster composition and Disease stage composition are updated with the new cluster IDs as shown in Fig. R7G.
- (d) Fig. 3B,C, reproduced here as Fig. R8B,C, is updated with new confusion matrices for their respective comparison as shown in Fig. R9B,C.
- (e) Fig. 3E, reproduced here as Fig. R8E, is updated with the new barplot showing GINI importance score for prediabetes vs diabetes comparison as shown in Fig. R9E.
- (f) Fig. S9 here reproduced as Fig. R4 from the previous version of the manuscript is replaced with Fig. S10 here reproduced as Fig. R5, with evaluations calculated via new cluster enrichment vectors per individual with updated prediction labels for C_{VAE} clusters.
- (g) Fig. S10 here reproduced as Fig. R10 from the previous version of the manuscript is replaced with Fig. S11, here reproduced as Fig. 9, with evaluations calculated via new cluster enrichment vectors per individual with updated prediction labels for C_{HFE} clusters.
- (h) Additional 2-way comparisons introduced in Fig. S10 from the previous version of the manuscript, reproduced as Fig. R10, were validated and a new supplementary figure is added as Fig. S12 in the revised manuscript and shown below as Fig. R12 with updated predictions for different disease cohorts using C_{HFE} clusters.

We have added the details of the consensus clustering algorithm in the methods section on line 518 of the revised manuscript.

Previous version of clustering algorithm on line 399 of the methods section

- (a) **Feature selection and graph construction:**
We selected the VAE-obtained features from the subsampled, well-represented dataset $B_{i,d}$, where i represents individual samples, and d represents disease conditions. A graph was then constructed based on neighborhood connections, calculated using the Euclidean distance between VAE-obtained features.

Figure R9: (corresponding to the updated Fig. 3 in the revised manuscript)

(b) **Neighborhood and clustering:**

We set the neighborhood size to 10 and applied Leiden clustering algorithm on the graph. This clustering resulted in 9 distinct clusters, denoted as C_z , where $z = 0, 1, \dots, 8$.

Revised version of clustering algorithm on line 523 of the revised manuscript.

Consensus Leiden Clustering

To obtain robust cluster label assignments, we applied a consensus clustering strategy (Monti et al., 2003) using the Leiden algorithm (Traag et al., 2019) as the base partitioner. First, a k-nearest neighbor (kNN) graph was constructed from the cell-feature matrix using Euclidean distance and a fixed neighborhood size ($k = 10$, unless stated otherwise). We then performed $n = 30$ independent Leiden runs with varying random seeds at resolution parameter 1.0. For each run, the cluster assignments were recorded, and a co-association matrix was built, where entry (i,j) reflects the proportion of runs in which cells i and j were assigned to the same cluster.

The co-association matrix was subsequently converted into a dissimilarity matrix ($D = 1 - P$), which was subjected to average-linkage hierarchical clustering. The final consensus partition was derived by cutting the dendrogram at a threshold corresponding to a minimum co-association of 0.8, thereby retaining only densely supported clusters.

To assess robustness, we computed the Adjusted Rand Index (ARI) between the consensus partition and each individual Leiden run. Reported stability scores include the mean and standard deviation of ARI across runs. This procedure ensures that the reported clusters represent stable and reproducible communities, rather than artifacts of stochastic initialization. We iteratively run the above algorithm by removing the clusters with fewer than 30 cells until the sizes and number of clusters converge, and/ or we obtain the mean $ARI \geq 0.8$.

Figure R10: (corresponding to the old Fig. S10) Handcrafted Features: (A) Heatmap where each row indicates the enrichment vector of nuclei from an individual across different cluster labels. Each row head annotations indicate the (left) ground truth and (right) predicted disease stage of T2DM. The color map indicates the fraction of nuclei from PBMCs of individual samples in a particular cluster. (B) Disease stage-specific enrichment profile, $Enrichment_d$ (see Methods) across different clusters (C) Cluster-specific enrichment profile, $Enrichment_h$ (see Methods) across different stages of T2DM. The 2×2 confusion matrix indicates the 2-way classification of individual samples from (D) prediabetic–diabetic, (E) healthy–diabetic, and (F) healthy–prediabetic individuals upon leaving one out cross-validation for RFC trained on the enrichment profile $Enrichment_i$ of remaining samples.

Figure R11: (corresponding to the updated Fig. S11 in the revised manuscript) Handcrafted feature and embedding; Heatmap where each row indicates enrichment vector of nuclei from an individual from (A) All three pathologies (B) Healthy vs Prediabetic, (C) Diabetic vs Prediabetic, and (D) Healthy vs Diabetic across different cluster labels. Each row head annotations indicate the (left) ground truth and (right) predicted pathology. (E) 5-fold cross-validation averaged prediction of Pathology-wise and overall recall against random permutation of Pathology labels as Null (iter = 200). (F) 5-fold cross-validation averaged True positive rate of Pathology binary labels against random label permutation as Null (iter = 200). (G) 3x3 confusion matrix showing LOOCV prediction rates for different pathologies. (H) ROC curve representing false positive probability vs true positive probability of binary class label for different pathologies, and overall true positive probability vs false positive probability.

Figure R12: (corresponding to the new Fig. S12 in the revised manuscript) Confusion matrix showing TPR for LOOCV two-way classification between (A) Healthy vs Prediabetic (B) Prediabetic vs Diabetic (C) Healthy vs Diabetic. ROC curves showing Probability for False Positive prediction vs True positive prediction with 5-fold CV for (D) Healthy vs Diabetic, (E) Diabetic vs Prediabetic, and (F) Diabetic vs Healthy individuals. Bar plot indicates the two-way prediction recall of (G) Healthy vs Prediabetic (H) Prediabetic vs Diabetic (I) Healthy vs Diabetic, averaged over 5-fold cross-validation (blue) against shuffled randomized label permutation (orange). The black annotation on the bar plot marks 1 standard deviation. (J) Cluster GINI importance score averaged over LOOCV 2-way prediction between different pathologies. (K), (L), (M), and (N) Box plot representing Enrichment score calculated for Clusters 13, 5, 11, and 4 of C_{HFE} across different pathologies. Each dot represents an individual.

-
10. Authors have selected CD25 as PBMC activation marker. Although CD25 is a known marker for activation, have you tested other activation markers such as CD69, HLA-DR to confirm your findings?
-

Author reply: A previous study [4] found increased expression of CD25+ activated T cells in T2DM due to chronic inflammation. This is especially true in individuals with T2DM along with other comorbidities, where there is increased expression of both CD69 and CD25 [25]. While CD69 activation is associated with a specific subset of T cells [29], we were not able to find conclusive studies that demonstrate pan-activation of immune cells. Also, as stated in previous studies [45, 41], HLA-DR showed lower expression in monocytes in T2DM. CD69, CD25, and HLA-DR also have specific activation timescales [39]. While CD69 is considered an early activation marker (a few seconds to hours), CD25 expression response has an intermediate timescale of a few hours to days, whereas HLA-DR expression is a late stage activation marker expressed after a few days [1, 39]. Given these considerations, for our observational study of unlabelled peripheral blood samples, we selected CD25 as activation marker to analyze some of the differences we see in chromatin states of pan-immune cells, and we did not explore other markers.

11. Fig S12C and Fig 5: how higher heterochromatin/H3K9Me3/gene silencing is associated with more activation? What could be the mechanism? Is it that CD25 is probably indicating a specific subset of immune cells rather than presenting an overall activation?
-

Author reply: We acknowledge the reviewer’s insightful question regarding how higher heterochromatin silencing may be associated with increased activation. Earlier work has demonstrated that a high-glucose environment leaves persistent chromatin “memory” with loss of repressive H3K9me3 and gain of activating marks, thereby sustaining NF- κ B/p65 expression and downstream inflammatory gene activity [47]. In diabetic monocytes, increased H3K9me2/3 at specific loci such as IL1A and PTEN correlates with transcriptional shifts toward inflammation [30], while downregulation of the lncRNA DRAIR enhances G9a-mediated H3K9me2/3 at anti-inflammatory targets (e.g., IL1RN), effectively silencing them and sustaining a pro-inflammatory phenotype [40]. Importantly, these epigenetic modifications can be inherited: recent work shows that hyperglycemia exposure in CD34+ hematopoietic progenitors induces senescent features and epigenetic remodeling (reduced H3K9me3 at the NF- κ B-p65 promoter and gain of activating marks) that persist in their myeloid progeny, which display elevated SASP cytokine secretion and inflammatory responsiveness [48]. Together, these findings suggest that heterochromatin accumulation at regulatory/anti-inflammatory loci removes negative feedback, thereby favoring heightened immune activation despite overall increases in H3K9me3. We also concur with the reviewer that CD25 expression is unlikely to reflect global PBMC activation, but more likely indicates expansion or activation of specific immune subsets driven by the chronic low-grade inflammation characteristic of T2DM.

Revisions: **We have discussed this point on line 272 of the revised manuscript.** We also find an increase in heterochromatin and an association of activation with higher

heterochromatin. These results support our analysis on Hoechst-stained images, which identified chromatin remodeling-associated features (e.g., condensed chromatin loci, nuclear size, and intensity distribution) as important in differentiating disease stages of T2DM. This is in line with previous studies showing that in T2DM, hyperglycemia induces persistent epigenetic remodeling—such as altered H3K9me3 at inflammatory and regulatory loci in endothelial cells, monocytes, and progenitors—that silences anti-inflammatory brakes while sustaining NF- κ B activity, thereby favoring subset-specific (e.g., CD25+ T cell) rather than global PBMC activation [47, 30, 40, 48].

12. Fig 5: I wonder why the data for Lamin, H3K9Me3 and CD25 in prediabetes not presented?
-

Author reply: We focused on characterizing the largest effect attributable to diabetes, and therefore selected non-glycemic healthy individuals as the control group. Individuals with diabetes demonstrated the strongest specificity and sensitivity in our analyses (Fig. R4E,F), whereas healthy individuals showed no specificity, justifying this choice of comparator. In the unlabeled, pan-PBMC analysis, we lack a priori definitions for the correct subdivision of immune cell types. Our results suggest that the prediabetic group exhibits substantial subpopulation stratification. To avoid confounding comparisons across heterogeneous baselines, we restricted our primary contrast to healthy versus diabetic samples for our immunofluorescence experiments.

13. Methods: Should include criteria of defining prediabetes and T2D.
-

Author reply: The initial diagnosis was performed using a combination of gold standard clinical methods such as fasting Plasma Glucose (FPG), oral glucose tolerance test (OGTT), and glycated hemoglobin (HbA1c). We provide the primary, secondary, and tertiary diagnosis for each individual in Table 1.

Revisions: **We revised the methods section of the manuscript on line 345:**

The initial diagnosis was performed using a combination of gold standard clinical methods such as fasting Plasma Glucose (FPG), oral glucose tolerance test (OGTT), and glycated hemoglobin (HbA1c). We provide the primary, secondary, and tertiary diagnoses for each individual in Table 1.

14. Methods: use of rabbit anti- γ -H2AX is mentioned for study of DNA damage. However, I do not find any data for this.
-

Author reply: We apologize for this oversight during the editing of our manuscript. We have replaced that sentence with the correct description of our immunostaining panel with the correct catalog number and company name. For immunofluorescence experiments of PBMCs, Monoclonal mouse anti-Lamin A (catalog no. ab8980, Abcam); Rabbit

Polyclonal Anti-Histone H3 trimethylation K9 (catalog no. ab8898, Abcam); Mouse Monoclonal IL2RA Antibody (catalog no. AB6411, Abcam) was used.

Revisions: **We edited the sentence to what appears on line 452-455:**

For immunostaining of PBMCs, Monoclonal mouse anti-Lamin A (catalog no. ab8980, Abcam); Rabbit Polyclonal Anti-Histone H3 trimethylation K9 (catalog no. ab8898, Abcam); Mouse Monoclonal IL2RA Antibody (catalog no. AB6411, Abcam) was used.

15. Discussion should include thoughts on the top important features identified to be segregating disease states and how they might be linked/changed with disease states.
-

Author reply: We agree with the reviewer, and we have added the details of the top features, particularly descriptors of condensed chromatin loci, nuclear size, and intensity distribution. These imaging-derived features are consistent with chromatin remodeling events known to accompany metabolic dysfunction. Specifically, we observed an overall increase in heterochromatin content, decreased Lamin expression, and a higher fraction of PBMC activation in T2DM. Such processes are thought to favor subset-specific activation (e.g., CD25+ T cells) rather than global PBMC activation [47, 30, 40, 48]. Taken together, our findings suggest that the image-derived chromatin features can capture aspects of the epigenetic alterations that have been implicated in the immune dysregulation observed in the progression of T2DM.

Revisions:

We added the following statement on line 264 of the revised manuscript:

Quantitative descriptors capturing loci of condensed chromatin, nuclear size, and intensity distribution were the key features that enabled identification of the enriched PBMC subpopulation specific to diabetes and prediabetes.

We added the following statement on line 272 of the revised manuscript:

We also find an increase in heterochromatin and an association of activation with higher heterochromatin. These results support our analysis of Hoechst-stained images, which identified chromatin remodeling-associated features (e.g., condensed chromatin loci, nuclear size, and intensity distribution) as important in differentiating disease stages of T2DM.

16. Authors need to discuss study strengths, but more so the limitations including validation of their algorithms in an independent cohort, associations with other co-morbidities, diabetes complications (if known), lack of immune cell analyses, functional analysis of the immune cells, etc.
-

Author reply: In the revised manuscript, we have expanded our Discussion to highlight both the strengths and limitations of our study. Specifically, we acknowledge that our algorithms have not yet been validated in an independent cohort, and that future multi-center studies will be necessary to establish generalizability. Furthermore, while our

chromatin imaging approach could provide a functional readout of peripheral immune cell dysregulation, we did not perform complementary immune cell subpopulation profiling or mechanistic assays (e.g., FACS-based profiling, T cell activation assays) that would directly link chromatin architecture to PBMC function. These points are now explicitly discussed in the revised manuscript.

Revisions: **We have added a paragraph highlighting limitations of this study on line 326 of the revised manuscript:**

Nevertheless, several limitations should be noted. For instance, our algorithms were developed and tested in a single cohort, and independent validation in larger, multi-center cohorts will be required to establish generalizability. The dependence on research lab infrastructure (e.g. centrifuge, confocal microscope, workflow complexity), can be addressed by developing specialized automated imaging and screening workflows. Moreover, our approach does not provide detailed immune cell subpopulation resolution, nor did we perform functional assays (e.g., cytokine secretion, T cell or macrophage activation studies) to mechanistically link chromatin architecture to immune dysfunction. Future studies integrating chromatin imaging with single-cell sequencing or proteomics will be critical to further validate and extend our findings.

Minor Comments:

17. Title and abstract should reflect the actual message. Please modify.
-

Author reply: We have changed the abstract and the title of the manuscript to better reflect the findings of our manuscript.

Revisions:

Revised Title: Detecting chromatin state alterations in PBMCs associated with Type 2 Diabetes Mellitus

Revised Abstract: Type 2 Diabetes Mellitus (T2DM) involves patient-specific immune dysfunction not addressed by current diabetes management methods. Traditional methods to assess functional changes to peripheral blood mononuclear cells (PBMCs) are costly and time-intensive for routine use. Here, we present an orthogonal and adjunct chromatin imaging-based assay to monitor alterations to PBMCs during T2DM progression. We studied 57 individuals across healthy, prediabetic, and diabetic stages, performing chromatin imaging of live PBMCs using a microfluidic imaging assay. By applying representation learning on chromatin images, we identified distinct PBMC clusters, with specific subpopulations enriched at different T2DM stages. We found that the levels of certain PBMC subsets are predictive of T2DM. Additionally, we observed significant changes in nuclear and chromatin mechanics in diabetic individuals compared to prediabetic and healthy individuals, along with decreased Lamin A/C expression and increased cellular activation in diabetic PBMCs. Collectively, this study provides a cost-effective and scalable solution for the routine monitoring of T2DM using PBMC chromatin biomarkers.

18. Tracking inflammation during the T2DM progression with sequencing and proteomics techniques remains costly, labor-intensive, and impractical”. Provide a reference.
-

Author reply: We have modified the statement and added the references to the above text.

Revisions: **We have revised the text that appears on line 59 in the revised manuscript to reflect these changes.**

While tracking immune cell subpopulation stratification can be used to monitor T2DM, this requires methods such as single-cell sequencing [28] and proteomics [44] which remain too costly, labor-intensive, and impractical for a clinical setting.

19. Intro: “In summary, we demonstrate an affordable, compact, scalable, point-of-care (POC) chromatin imaging assay to monitor immune dysfunction in T2DM management for personalized disease tracking.” I think this is an overstatement. Change and reflect what exactly is shown in this paper.
-

Author reply: We acknowledge the reviewer’s feedback on the phrasing of our concluding statement in the Introduction and have modified it to align with their suggestion.

Revisions:

We have changed the statement on line 50 in the previous version of the manuscript to the following on line 73 in the revised version of the manuscript:

In summary, we demonstrate that our chromatin imaging assay can be an affordable and scalable approach for potential clinical use in monitoring PBMC chromatin alterations relevant to T2DM management and personalized disease tracking.

20. “Although chromatin images from individual T2DM stages looked . . . cluster 6 showed enrichment in diabetic individuals (Fig. S6A, S6B)”. The reference of figures is incorrect here.
-

Author reply: We thank the reviewer for pointing this out. In the revised version of the manuscript, we have demonstrated this using box plots in Fig. R5 K,L, and we find a statistically significant difference in enrichments of supopulations of prediabetic and diabetic individuals.

Revisions:

We have corrected the reference to the supplementary figure Fig. S6A,B on line 129 in the previous version to more appropriately refer to Fig. S10K,L on line 184 in the revised version of the manuscript.

-
21. Discussion: Ref 25 is very specific to a particular region and the statement “Inflammatory biomarkers such as C-reactive protein (CRP), (IL6) show significant heterogeneity across patients [25]” would need more global evidence.
-

Author reply: We agree that the cited study is specific to a region. In the revised manuscript, we have added references to studies conducted over intercohort data and other geographical locations. In particular, the cited study conducted at a different geographical location shows that readouts such as CRP and IL-6 are not specific and associated with increased risk of T2DM, for instance CRP: 1.5–12.0 (rel. risk = 4.2); IL-6 Range: 0.9–5.6 (rel.risk = 2.3) (e.g., CRP relative risk 4.2, IL-6 relative risk 2.3) with significant overlap with control population [37]. These ranges of biomarker readout are in line with the previously cited study [36]. Moreover, multivariate models developed over inter-cohort data show that CRP readouts along with other biomarkers (such as adiponectin, ferritin, interleukin-2 receptor A (IL2RA), glucose, and insulin) achieve accuracies around 76–78% in predicting at-risk patients [29].

Revisions:

We have added the following text on line 310 of the revised manuscript:

This is similar to currently available immune-related tests for detecting diabetes, including HbA1c, CRP, and IL-6. HbA1c with well-validated cut-offs ($\text{HbA1c} \geq 6.5\%$) show high specificity ($\geq 90\%$), but moderate sensitivity (45-70%) [19, 26, 3]. CRP and IL-6 are established inflammatory markers associated with increased risk of T2DM (e.g., CRP relative risk 4.2, IL-6 relative risk 2.3) [37], and in multivariate models combined with other biomarkers (such as adiponectin, ferritin, interleukin-2 receptor A (IL2RA), glucose, and insulin) achieve accuracies around 76–78% in predicting at risk patients [29] (Table S3).

Reviewer 2 (Remarks to the Author):

This study presents a novel imaging-based assay to detect immune alterations across Type 2 Diabetes Mellitus (T2DM) stages by profiling chromatin architecture in live PBMCs. By integrating 3D chromatin imaging with a variational autoencoder and clustering, the authors identify nine chromatin states differentially enriched across healthy, prediabetic, and diabetic individuals. These chromatin features, along with nuclear mechanical properties and marker expression, support the potential of chromatin structure as a functional immune biomarker. While the methodology is technically innovative and the assay is proposed as scalable, its clinical applicability is limited by small sample size, lack of immune cell-type validation, and infrastructural complexity. A revision to address these issues will improve this work before publication.

Author reply: We thank the reviewer for their constructive feedback. We agree that the small sample size is a limitation, and we have highlighted this in the revised Discussion. To strengthen our analysis, we added validation using an independent test set, k-fold cross-validation (CV), and permutation testing. We now also report confidence intervals and ROC threshold analysis to provide interpretable sensitivity/specificity trade-offs. These additions demonstrate that our results are robust within the current cohort. Immune cell-type classification and infrastructural optimization are beyond the scope of the present manuscript but represent interesting directions for future work.

Detailed answers to the reviewer’s comments are provided below. To answer some of the comments, we have used the analysis provided to reviewer 1 and referenced the figures in those responses.

Major comments:

1. While the model demonstrates promising classification accuracy, it currently functions as a black-box predictor without interpretable or actionable thresholds. For clinical relevance, the authors should consider providing calibrated probability scores, ROC curves, or defining rule-based chromatin signatures (e.g., percent of cells in Cluster 6) that can be used to guide diagnostic decisions. Can the authors clarify whether such thresholds were explored or could be derived from their enrichment features?
-

Author reply: We thank the reviewer for this constructive suggestion. We have incorporated ROC curves for 5 Fold CV comparison in Fig. S10H,I in the revised manuscript, shown as Fig. R5H,I above. We find higher specificity in the classification of Prediabetic and Diabetic individuals if all the pathology states exist in the population. Notably, in the new Fig. S12D-F in the revised manuscript (shown as Fig. R12D-F above), we find an AUC of 0.77 for Healthy vs Prediabetic, 0.85 for Diabetic vs Prediabetic, and 0.65 for Healthy vs Diabetic. We have also incorporated the pathology-specific cluster enrichment analysis for both those that were identified from VAE feature embedding C_{VAE} and handcrafted feature embedding C_{HFE} . We indeed find pathology-specific enrichment of specific clusters. Notably, in the C_{HFE} , Cluster 5 shows enrichment in Healthy and Diabetic, whereas Cluster 4, 11, and 13 show enrichment in Prediabetic individuals, as shown in the new Fig. S12K-N, which is shown as Fig. R12K-N above. The difference in enrichment of Cluster 5 in Diabetic and Healthy individuals is significant. As stated

in the previous version of the manuscript, we also find in C_{VAE} , Cluster 6 is enriched in Prediabetic individuals and Cluster 1 is enriched in Healthy and Diabetic individuals, as shown in Fig. S10K,L in the revised manuscript, which is shown as Fig. R5K,L above. These pathology-specific differences are statistically significant after FDR correction using Benjamini-Hochberg correction. Based on these enrichment scores, the pathology-specific thresholds can be evaluated. Moreover, to estimate the performance of our method on a general cohort, we have also evaluated the sensitivity and specificity of prediction against a predefined target on an independent test set. We have summarized these findings in Table 2.

Revisions: **As described above, we have replaced the previous Fig. S9 and S10 with updated Fig. S10, S11, and a new Fig. S12. In addition, we added the following table.**

Class	Th	Sens	Sens CI Low	Sens CI High	Spec	Spec CI Low	Spec CI High	ROC AUC OvR	PR AUC OvR
Diabetic	0.293	0.75	0.194	0.994	0.778	0.400	0.972	0.833	0.710
Prediabetic	0.577	0.75	0.194	0.994	0.556	0.212	0.863	0.694	0.586
Healthy	0.433	0.20	0.005	0.716	0.875	0.473	0.997	0.575	0.469

Table 2: Per-class performance metrics with thresholds, confusion counts, sensitivity/specificity (with CIs), and One vs Rest (OvR) AUCs. for independent test set. Sensitivity for Diabetes is 0.8, Prediabetes is 0.8, and Specificity target for Healthy is 0.80.

2. Was a power analysis conducted to justify the sample size (n=57)? While downsampling and cross-validation are applied, the authors do not statistically justify that the dataset is powered to detect meaningful inter-group differences.

Author reply: We did not perform a power analysis prior to the study. However, our retrospective analysis shows that with n=57 samples distributed across three groups (Diabetes:17; Prediabetes:22; Healthy:18), the study had 80% power to detect medium omnibus effects (Cohen’s $f \approx 0.43$, partial $\eta^2=0.15$), and large pairwise effects (Cohen’s $d \approx 0.9 - 1.0$). In the revised manuscript, we analyze the statistical significance for changes in subpopulation in the case of Diabetic vs Healthy and Diabetic vs Prediabetic, corrected for FDR in Fig. S10 K, L (here Fig. R5K,L) and Fig. S12K-N (here Fig. R12K-N).

3. The random forest classifier achieves AUCs $\geq 80\%$ and even 100% in some comparisons, raising concerns about overfitting in a small dataset. The manuscript would benefit from additional validation metrics (e.g., permutation tests, independent test set, confidence interval) to assess robustness and generalizability to larger or more diverse populations.

Author reply: We thank the reviewer for this important comment. In response, we have expanded this validation analysis to better assess the robustness and generalizability of the classifier. In the updated manuscript, we have performed permutation tests for each comparison and reported them alongside our performance evaluation metrics. We incorporated five-fold cross-validation in all prediction tasks to minimize potential biases. The resulting performance metrics are presented in Fig. S9 A (here R3 A), Fig. S10 E, F, J (here R5 E, F, J), Fig. S11 E, F (here R11 E, F) and Fig. S12 G, H, I (here R12 G, H, I). We find that the prediction outcome estimated by our model is significantly higher than the random chance and shows high sensitivity for diabetic and prediabetic

individuals. We next split the data into a training and an independent testing set before any preprocessing to avoid any transductive data leakages. Here, we used handcrafted features because these features can be independently obtained for each individual in the cohort. Next, we cluster the training data using Leiden clustering and assign the cluster labels to the testing data based on the nearest neighbor consensus. We assigned clusters to the testing data using KNN label transfer. We next evaluate the cluster enrichment vectors for each individual in the training and testing data. We find that the prediction performance of the model is in line with what we have found a priori. The results are shown in the new Fig. S13 in the revised manuscript (shown below as Fig. R13).

Figure R13: (corresponding to the new Fig. S13 in the revised manuscript) (A) Individuals sampled for training data, (B) Individuals held out from training data and their pathology prediction, (C) 3×3 Confusion Matrix showing prediction rates across different pathologies. (D) ROC curve representing the false positive rate for the expected true positive rate of the binary class label for different pathologies. 2×2 Confusion Matrix for classification on (E) Prediabetic vs Diabetic, (F) Healthy vs Diabetic, (G) Healthy vs Prediabetic.

Revisions: **We added the following details to the Results 3 section from line 172 - 198:**

Across different comparisons, the balanced accuracy of our model in 2-way classification tasks—distinguishing between diabetic vs. healthy is 0.58 from C_{VAE} and 0.72 from C_{HFE} with AROC of 0.65 as shown in Fig. 3B and Fig. S12A,D, validated against the null hypothesis by permutation testing as shown in Fig. S12G. Between diabetic vs. prediabetic individuals held-out from training the we find balanced accuracy of 84% with AROC of 0.70 from C_{VAE} and AROC of 0.85 with C_{HFE} as shown in Fig. 3C and Fig. S12B,E) validated against the null hypothesis by permutation testing as shown in Fig. S12H. The classification result of Healthy vs Prediabetic remained variable between the two approaches, but we see increased balanced accuracy of 0.7 and AROC of 0.77 with C_{HFE} against the null hypothesis, as shown in Fig. S12C,F,I, compared to C_{VAE} (Fig. S10G).

Although chromatin images from individual T2DM stages looked characteristically similar (Fig. 2B), cluster C_{VAE} 1 was significantly enriched in prediabetic patients (Fig. S10 K), while cluster C_{VAE} 6 showed significant enrichment in diabetic individuals (Fig. S10L) confirmed by the mean cumulative cluster importance score as shown in Fig. 3E. Validated by the mean GINI importance score in Fig. S12J. We derive similar but stronger enrichments using the C_{HFE} cluster, in particular for clusters C_{HFE} 4, 11, and 13, where we see significant enrichment of nuclei from individuals with prediabetes. Within cluster C_{HFE} 5, nuclei from diabetic individuals showed a significantly greater enrichment than healthy individuals, relative to the prediabetic individuals that show minimal enrichment. Notably, C_{HFE} 5 enriches for nuclei with maximum radial location of heterochromatin domains from the center “D3” and maximum separation of heterochromatin regions from the nuclear boundary represented by annotation “D4”, shown in Fig. S9J. These features were also identified as being discriminatory during disease state classification, Fig. 2F on a random training and test split, as shown in Fig. S9B. The three-way prediction between healthy, diabetic, and prediabetic individuals for both cluster enrichment features, we see higher recall and one vs rest AROC for prediabetic and diabetic individuals, as shown in Fig. S10E,F,H, and Fig. S11E-H.

We added the following section in the main manuscript on line 199:

Next we estimate the performance of our method on the general cohort using independent set analysis. To this end, we randomly split the training and testing sets before preprocessing into two separate cohorts of individuals, as shown in Fig. S13A,B. We then cluster and evaluate the enrichment vectors for training data as described previously. For testing data, we assign the cluster labels to the data using the KNN distance-based majority voting with training data as a reference to evaluate the enrichment vectors (see methods). Training on enrichment vectors for testing the data, we evaluated the prediction accuracy for the independent testing data. We find that the performance of the model is in line with the results observed in this study, with balanced accuracy of 75% and 77.5% Prediabetic vs Diabetic & Healthy vs Diabetic individuals at the default thresholds as shown in Fig. S13D,F. However, we saw significant confounding between Healthy and Prediabetic individuals, as shown in Fig. S13G. With a target sensitivity of 0.8 for Diabetic and Prediabetic individuals, and a target specificity of 0.8 for healthy individuals defined as the threshold for the training model, we find a specificity of 0.778 for Diabetic individuals for the independent set, as shown in Table 4 of the Supplementary Information.

We added the following section in the methods section on line 571:

Getting Individual specific cluster enrichment on independent set

We transferred cluster labels from a training split to an independent testing set using k -nearest neighbors (k-NN) voting in the reference feature space. A k-NN index ($k = 10$, cosine distance) was fit on the reference features. For each query sample, we retrieved its k nearest reference neighbors, tallied their labels, and used the normalized vote proportions as per-label probabilities; the predicted label is the arg max of these probabilities. Assignment confidence combines (i) the vote probability p for the predicted label and (ii) a distance-based compactness term. Let d be the query sample’s mean distance to its k neighbors. We estimate μ_{ref} and σ_{ref} as the mean and standard deviation of mean

neighbor distances computed on the reference set (excluding self-matches). Confidence is

$$\text{confidence} = \frac{1}{2} p + \frac{1}{2} \sigma \left(\frac{\mu_{\text{ref}} - d}{\sigma_{\text{ref}}} \right), \quad \sigma(x) = \frac{1}{1 + e^{-x}}.$$

Predictions with confidence below a tunable threshold (t) were rejected. The output reports the predicted label, vote probability, confidence, mean neighbor distance, a rejection flag, and per-label probabilities.

4. Chromatin condensation or nuclear changes are not specific to T2DM, they can also occur in response to aging, infection, cancer, or stress. The study does not establish mechanistic links between observed chromatin states and T2DM pathophysiology (e.g., beta-cell function, insulin resistance, or metabolic inflammation). The authors acknowledge that nuclear changes may reflect immune activation but don't tie them mechanistically to diabetes-specific processes.
-

Author reply: We agree with the reviewer's comment that the data presented is somewhat limited in establishing direct functional role between chromatin organizational changes with T2DM progression. To this end, we have updated the text that avoid over-interpretation of our results. We now emphasize that the observed decrease in Lamin expression, chromatin condensation, and activation in individuals with diabetes is consistent with the chromatin state differences identified in PBMC subpopulations through live chromatin imaging. In this way, our findings should be viewed as supportive of previous observations rather than conclusive evidence of causality.

Revisions: **We have modified the statement that appeared on line 178 in the previous version of the manuscript:**

Together, these observations provide a functional basis for the use of chromatin architecture as a biomarker to track T2DM-related immune alterations.

We have replaced this with the following statement on line 253 in the revised version of the manuscript:

These results reveal pronounced alterations in nuclear mechanics and chromatin condensation states of PBMCs in individuals with T2DM, suggesting that chromatin architecture could serve as a sensitive indicator of diabetes-associated immune dysfunction.

5. The identification of nine chromatin clusters is interesting, particularly the enrichment of Clusters 1 and 6 in specific disease stages. However, the biological significance of these clusters remains unclear. Are they associated with known PBMC subtypes (e.g., T cells, monocytes)? Flow cytometry or transcriptomic validation would considerably strengthen this link.
-

Author reply: We thank the reviewer for this insightful suggestion. Our primary aim in the present study is to show that chromatin imaging of peripheral blood cells is sensitive and specific to T2D-associated alterations, and that these image-derived features can *complement* established immune dysregulation biomarkers (e.g., IL-6, CRP). Because our

cohort does not include matched immunophenotyping or transcriptomic measurements, we cannot ascribe the nine chromatin clusters to canonical PBMC lineages (e.g., T cells, monocytes) with confidence. We view these experiments as a natural next step but beyond the scope of the current manuscript. We have clarified this limitation in the revised text.

Revisions: **We have added this text on line 326 of the revised manuscript:**

Nevertheless, several limitations should be noted. For instance, our algorithms were developed and tested in a single cohort, and independent validation in larger, multi-center cohorts will be required to establish reproducibility and generalizability. The dependence on research lab infrastructure (e.g. centrifuge, confocal microscope, workflow complexity), can be addressed by developing specialized automated imaging and screening workflows. Moreover, our approach does not provide detailed immune cell subpopulation resolution, nor did we perform functional assays (e.g., cytokine secretion, T cell or macrophage activation studies) to mechanistically link chromatin architecture to immune dysfunction. Future studies integrating chromatin imaging with single-cell sequencing or proteomics will be critical to validate and extend these findings.

6. How does the proposed chromatin imaging assay compare to established biomarkers like HbA1c, CRP, or IL-6 in terms of diagnostic power, cost, ease of use, and scalability? Can the authors discuss its integration into current clinical workflows or its potential complementarity?
-

Author reply: We agree with the reviewer that a comparative analysis will be helpful in placing our results in the context of established diagnostic approaches. When applied to our cohort, our approach achieved a specificity of 0.78 (CI: 0.4 - 0.972) and sensitivity of 0.75 (CI: 0.194 - 0.994) in distinguishing diabetic individuals. In this context, HbA1c with well-validated cut-offs (HbA1c $\geq 6.5\%$) showed high specificity ($\geq 90\%$), but moderate sensitivity (45-70%)[19, 26, 3]. CRP and IL-6 are established inflammatory markers associated with increased risk of T2DM (e.g., CRP relative risk 4.2, IL-6 relative risk 2.3)[37], and in multivariate models combined with other biomarkers (for eg. adiponectin, ferritin, interleukin-2 receptor A (IL2RA), glucose, and insulin) achieve accuracies around 76–78% in predicting at risk patients [29] (Table S3). From a cost perspective, HbA1c and CRP are low-cost assays [15, 22], while IL-6 is substantially more expensive [8] (Table S3). Our approach remains inexpensive and uses low-cost reagents such as Hoechst dye and microfluidics devices [21], suggesting material costs could be comparable with scalability (Table S3). The current limitation requiring research lab infrastructure (e.g. centrifuge, confocal microscope, workflow complexity) can be addressed with automation and scalability. Importantly, our approach provides complementary information about the functional state of the immune cell that is not directly addressed by established biomarkers. Taken together, our approach has the potential to integrate into centralized lab workflows as an adjunctive assay, providing a sensitive, orthogonal readout of immune dysregulation in T2DM.

Revisions: **We have added the following sentences to the discussion section on line 308 of the revised manuscript:**

When applied to our cohort, our approach achieved a specificity of 0.78 (CI: 0.4 - 0.972) and sensitivity of 0.75 (CI: 0.194 - 0.994) in distinguishing diabetic individuals. In this context, HbA1c with well-validated cut-offs (HbA1c \geq 6.5%) show high specificity (\geq 90%), but moderate sensitivity (45-70%)[19, 26, 3]. CRP and IL-6 are established inflammatory markers associated with increased risk of T2DM (e.g., CRP relative risk 4.2, IL-6 relative risk 2.3)[37], and in multivariate models combined with other biomarkers (for eg. adiponectin, ferritin, interleukin-2 receptor A (IL2RA), glucose, and insulin) achieve accuracies around 76–78% in predicting at risk patients [29] (Table S3). From a cost perspective, HbA1c and CRP are low-cost assays [15, 22], while IL-6 is substantially more expensive [8] (Table S3). Our approach remains inexpensive and uses low-cost reagents such as Hoechst dye and microfluidics devices [21], suggesting material costs could be comparable with scalability (Table S3). The current limitation requiring research lab infrastructure (e.g., centrifuge, confocal microscope, workflow complexity) can be addressed with automation and scalability. Importantly, our approach provides complementary information about the functional state of the immune cell that is not directly addressed by established biomarkers.

We have included the following table in the Supplementary section of the Revised manuscript:

Dimension	HbA1c	CRP / IL-6	Chromatin imaging (this work)
Primary signal	Long-term glycemia (2-3 months) [3], non indicative of patient specific inflammation	Systemic inflammation [29]	Alteration in chromatin organization in PBMCs due to sub-population stratification, activation or senescence
Clinical role	Diagnostic & monitoring standard	Adjunct risk / inflammation marker	Exploratory/adjunct; Orthogonal to established methods
Diagnostic power	Established cut-off: HbA _{1c} \geq 6.5%; high specificity(\geq 90% [19]); Moderate sensitivity (45.5%[19],70%[26])	Associated with risk but non-specific [37, 29] CRP: 1.5–12.0 (rel. risk = 4.2); IL-6 Range: 0.9–5.6 (rel. risk = 2.3) [37], multivariate accuracy 76-78% [29]	For our cohort, specificity of 0.78 (CI: 0.4 - 0.972) and sensitivity of 0.75 (CI: 0.194 - 0.994) in distinguishing diabetic individuals.
Cost per test (relative)	Low (\$8 – \$16) [15]	CRP: Low (\$12 – \$16)[22]; IL-6: Moderate: (\$100 - \$300) [8]	Low (Estimated with Scalability): \leq \$10; includes low cost reagents like Hoechst (\$143/10 mL Thermo Fischer) & Microfluidics Devices [21]
Capital/equipment	Standard clinical analyzer	Standard immunoassay platforms	Benchtopy centrifuge + density-gradient separation + confocal microscope (or high-NA wide-field) + compute for ML
Ease of use	High	High	automation feasible
Throughput / TAT	High / same-day	High / hours	Moderate (same-day feasible); automation can increase throughput
Scalability	Mature	Mature	Scalable with plate-based imaging, automated segmentation, model deployment; not POC yet
Complementarity	Glycemic exposure	Inflammatory burden	Promising sensitive and specific approach in the context of this cohort size; orthogonal to established methods.

Table 3: Comparison of HbA1c, CRP/IL-6, and chromatin imaging assay across diagnostic and practical dimensions.

Minor comments:

- The term "point-of-care" is used throughout the manuscript. However, the workflow

still involves confocal imaging and machine learning, which may not yet be feasible in decentralized clinical settings. Clarify whether this term refers to future potential or current readiness.

Author reply: We agree with the reviewer’s comment. Our current workflow does rely on research infrastructure, including confocal imaging, centrifugation, and machine learning algorithms, and is therefore not directly translatable as a point-of-care device at this stage. We acknowledge that in future work, we will adapt each of these components for true point-of-care applicability. To avoid overstating the current readiness, we have removed the term ‘point-of-care’ from the manuscript.

Revisions: We have made the following changes to the previous version of the manuscript to reflect this:

Previous version at line 6: Type 2 Diabetes Mellitus (T2DM) involves patient-specific immune dysfunction not addressed by current point-of-care (POC) technologies.

Present version at line 6: Type 2 Diabetes Mellitus (T2DM) involves patient-specific immune dysfunction not addressed by current diabetes management devices

Previous version at line 8: Here we present a chromatin imaging-based POC assay to monitor immune dysfunction during T2DM progression.

Present version at line 8: Here we present a chromatin imaging-based assay to monitor PBMC subpopulation stratification during T2DM progression

Previous version at line 21: While recent advances in point-of-care (POC) technology have enhanced the accuracy and miniaturization of diabetes management tools such as insulin delivery systems and blood glucose monitors [2, 9], current POC devices do not address patient-to-patient variability in Type 2 Diabetes Mellitus (T2DM) related immune dysfunction.

Present version at line 20: While recent advances in diabetes care and management technology have enhanced miniaturization and accessibility, such as insulin delivery systems and blood glucose monitors [2, 9], they do not address patient-to-patient variability in Type 2 Diabetes Mellitus (T2DM) related immune dysfunction.

Previous version at line 70: In summary, we demonstrate an affordable, compact, scalable, point-of-care (POC) chromatin imaging assay to monitor immune dysfunction in T2DM management for personalized disease tracking.

Present version at line 72: In summary, we demonstrate that chromatin imaging assay can be an affordable, compact, and scalable approach for potential clinical use in monitoring PBMC subpopulation stratification relevant to T2DM management and personalized disease tracking.

8. The authors mention future release of code and data. Please specify a timeline or repository (e.g., GitHub) to ensure reproducibility.
-

Author reply: We have uploaded the code used in our analysis to GitHub at the following link. In addition, the data supporting the analyses have been deposited and are available

for reproducibility. Together, these resources ensure that all steps of our workflow can be reproduced and extended by others.

Reviewer 3 (Remarks to the Author):

Dr. Afarani et al. present a novel assay using chromatin imaging to monitor immune dysfunction in T2D. They studied 57 individuals across healthy, prediabetic, and diabetic stages, using microfluidic imaging of live PBMCs. Through representation learning on chromatin images, they identified nine PBMC clusters, with some PBMC subsets predictive of T2DM. Diabetic PBMCs exhibited altered nuclear and chromatin mechanics, decreased Lamin A/C expression, and increased cellular activation. The approach is interesting and potentially of use in different contexts of disease monitoring. However, I have a number of concerns:

Author reply: We sincerely thank the reviewer for their careful evaluation of our work and for the constructive feedback provided. We appreciate the recognition of the novelty and potential clinical relevance of our chromatin imaging approach, and we value the specific concerns raised. We have carefully revised the manuscript to address these points, clarified our interpretations, and incorporated additional discussion where appropriate. We believe these revisions strengthen the manuscript and improve its clarity for readers.

Detailed answers to the reviewer's comments are provided below. To answer some of the comments, we have used the analysis provided to reviewer 1 and referenced the figures in those responses.

Introduction:

1. The authors state that tissue alterations in the adipose tissue microenvironment send “chemical signals” into the bloodstream, influencing the function of other organs. This concept requires elaboration and mechanistic detail. Please expand on how adipose tissue-derived factors (e.g., adipokines, cytokines, exosomes) influence systemic immune alterations (doi: 10.1016/j.isci.2024.110528).
-

Author reply: In the revised manuscript, we have added a description of the functional basis for alterations in peripheral blood immune cell subpopulations during T2DM, driven by hyperglycemia, elevated free fatty acids, and pro-inflammatory cytokines [42]. In addition, we have incorporated mechanistic details on adipose tissue-derived cytokines and their effects on the liver, pancreas, and kidneys, highlighting their role in sustaining chronic low-grade inflammation as demonstrated in previous studies [49, 18, 50]. The edited version of the introduction highlighting these changes is provided below. We like to thank the reviewer for pointing us to the relevant literature, we have added these references to the revised version of the manuscript.

Revisions: **We have revised the following sentences in the introduction on line 24 – 32 of the older version of the manuscript:**

Local alterations in the adipose tissue microenvironment during the onset of T2DM send

chemical signals into the bloodstream, influencing the function of various organs — primarily affecting the pancreas, liver, and kidneys [35, 7, 20]. Key soluble signals, including altered glucose levels, chemokines, and cytokines like TNF- α , IL-6, and IL-1 β , contribute to the systemic metabolic dysregulation associated with T2DM [38]. These signals impair immune cell function, leading to immune dysfunction, a hallmark feature of T2DM progression [42].

Above text has been replaced with a more comprehensive introduction that appears on line 23 – 43 in the revised version of the manuscript:

In T2DM, the inflammatory environment—characterized by hyperglycemia, elevated free fatty acids, and cytokines—pushes immune cells toward glycolytic, pro-inflammatory phenotypes, resulting in subpopulation imbalance [56, 52, 11, 16, 27]. For instance, T2DM onset is associated with the accumulation of pro-inflammatory T cell subsets, such as Th1 and Th17 cells, and a reduction in anti-inflammatory regulatory T cells in the adipose tissue due to altered insulin signalling [42]. Changes in circulating and resident immune cell subpopulations in the adipose tissue during the onset of T2DM drive compositional remodeling of the visceral adipose tissue (VAT) microenvironment skewing resident macrophages toward a pro-inflammatory M1 phenotype, which secrete cytokines such as TNF- α , IL-6, and IL-1 β [20]. Together with infiltrating subpopulations of NK and B lymphocytes [17, 51], these M1 macrophages amplify inflammation in microenvironment of VAT and other organs, leading to patient specific comorbidities [35, 7].

In the liver, adipose-derived cytokines and immune cell trafficking activate Kupffer cells and recruit monocytes and T cells, promoting NASH and fibrosis in certain patient cohorts [49]. Inclusive to that in specific patient subpopulations elevated levels of VAT-derived mediators (e.g., MCP-1, TNF- α , IL-6, IL-18) drive macrophage and T-cell infiltration into glomeruli and tubules, fueling chronic inflammation and fibrosis that characterize diabetic nephropathy [50]. Similarly, in pancreatic islets, macrophage-derived IL-1 β and TNF- α impair insulin secretion and reduce β -cell survival, linking adipose inflammation to endocrine dysfunction in T2DM [18].

-
2. Furthermore, partial exhaustion of tissue-infiltrating T cells has been demonstrated in T2DM and should be discussed (e.g., doi: [10.1172/jci.insight.139793](https://doi.org/10.1172/jci.insight.139793); doi: [10.1016/j.isci.2024.109032](https://doi.org/10.1016/j.isci.2024.109032)). Additionally, resistance of T cells to cytokine-mediated suppression is relevant (doi: [10.1136/bmjdr-2019-000772](https://doi.org/10.1136/bmjdr-2019-000772)).
-

Author reply: In the introduction of the revised manuscript, we have added details on functional changes in immune cells, including T cell senescence and immune activation, as reported in previous studies during the progression of T2DM [11, 43, 54, 24, 33]. This provides readers with the necessary context to interpret our compression assay findings, which revealed significant changes in the mechanical properties of immune cells. We thank the reviewer for suggesting the relevant literature in this area.

Revisions: **We have added the following sentences discussing this on line 43 – 53 in the revised version of the manuscript:**

Beyond shifts in immune cell subpopulation, the enhanced glycolytic activity and elevated ROS production observed in T2DM drive premature T cell senescence [11, 43]. The senescent T cells fail to effectively extravasate into target tissues [10, 43, 33] due

to cell-intrinsic metabolic defects contributing to surveillance deficits and secrete pro-inflammatory cytokines (like IFN- γ and TNF- α) contributing to the disease progression [54, 24]. Collectively, these findings suggest that altered and dysfunctional immune cell recirculation and local inflammatory cues from VAT integrate signals between metabolic organs and perpetuate disease progression. However, in this context, major gaps exist in detecting immune cell subpopulation stratification during the onset of T2DM in the clinically relevant setting.

3. The authors mention: “Recent multi-omic studies have identified significant shifts in leukocyte subpopulations and gene expression patterns throughout the progression of T2DM [8,9].” Please provide appropriate references to support this statement. It would be interesting to mention a multi-omics based approach in other diabetes subtypes (doi: 10.1038/s43856-025-00922-7)
-

Author reply: We have added more recent and appropriate list of references in the revised version of the manuscript [13, 34, 32, 23]. Since the studies evaluated the single-cell gene expression or selected palette of cell surface markers using FACS, we have modified the sentence to reflect that.

Revisions: **We modified the sentence as it appear on line 33 – 34 in the previous version of the manuscript:**

Recent multi-omic studies have identified significant shifts in leukocyte subpopulations and gene expression patterns throughout the progression of T2DM [5, 6].

We have replaced the above text with the following sentence on line 54–56 in the revised version of the manuscript:

Single-cell multi-omic studies of peripheral blood mononuclear cells (PBMCs) from T2DM patients versus healthy controls have unveiled shifts in immune cell subsets and activation states [13, 34, 32, 23].

4. The sentence: “Our findings indicate that specific PBMC subpopulations reveal patient-specific alterations in immune cells as T2DM advances.” is imprecise because there is no longitudinal follow-up of the same patients. Please rephrase this statement to reflect that differences were observed across groups, rather than progression within individuals.
-

Author reply: We thank the reviewer for pointing this out. We have now rephrased the wording to reflect that the presented study is a cross-sectional and not longitudinal.

Revisions: **We have made the following changes to reflect the above details:**

Previous version at Line 46: Our findings indicate that specific PBMC subpopulations reveal patient-specific alterations in immune cells as T2DM advances.

Revised version at Line 69 - 71: Our findings indicate that specific PBMC subpopulations reveal patient-specific alterations in immune cells in individuals with T2DM compared to prediabetic and healthy controls.

Results:

5. Diabetic and prediabetic groups show clearer separation (e.g., Fig. 2G), while discrimination of healthy individuals is less accurate. How do you explain this, and how might it be improved?
-

Author reply: We find that the differences between healthy and prediabetic individuals are less pronounced than those observed between prediabetic and diabetic cohorts. Recent single-cell multi-omics studies [13, 34, 32, 23] likewise do not emphasize marked functional differences between prediabetic and healthy groups. In contrast, a large bulk-sample study [12] reported differences in pathological profiles for both prediabetes and diabetes relative to healthy controls, in some cases even in opposite directions. Such discrepancies may reflect heterogeneity in the rate and trajectory of diabetes progression across individuals. As per your suggestion, we have improved our clustering approach and expanded the evaluation of the clustering methods, as shown in Fig. S9D,E (here Fig. R3D,E) for the clusters, C_{VAE} identified by VAE embedding, and in Fig. S9G,H (here Fig. R3G,H), C_{HFE} identified by handcrafted embedding. We find that the C_{HFE} was robustly clustered with ARI score = 0.91 (+/-) 0.01 (Fig. R3G,H) and the C_{VAE} is showing high degree of consensus across iterations with ARI score = 0.85 (+/-) 0.1 (Fig. R3D, E). We have reevaluated our classification performance using enrichments derived from an improved clustering approach on C_{HFE} , and we indeed find an improvement in the classification of healthy and diabetic individuals, with a balanced accuracy of 72%, as shown in Fig. S12 (here Fig. R12). These results have remained consistent with the performance of our model on an independent test set, yielding a balanced accuracy of 77.5% as shown in Fig. S13F (here Fig. R13 F). We want to highlight that the performance of the model on healthy individuals is less accurate, particularly because they are confounded with prediabetic individuals.

Revisions: **Specific updates to Fig. 2, Fig. 3, Fig. S10 (which was Fig. S9 in the previous version), and Fig. S11 (which was Fig. S10 in the previous version) are described below.**

- (a) Fig. 2A, reproduced above as Fig. R6A. The UMAP is updated with new cluster assignment labels as shown in Fig. R7A.
- (b) Fig. 2B, reproduced above as Fig. R6B. The representative images from each cluster were updated with new cluster IDs as shown in Fig. R7B.
- (c) Fig. 2G, reproduced above as Fig. R6G, The heatmaps showing Cluster composition and Disease stage composition are updated with the new cluster IDs as shown in Fig. R7G.
- (d) Fig. 3B,C, reproduced above as Fig. R8B,C, is updated with new confusion matrices for their respective comparison as shown in Fig. R9B,C.
- (e) Fig. 3E, reproduced above as Fig. R8E, is updated with the new barplot showing GINI importance score for prediabetes vs diabetes comparison as shown in Fig. R9E.
- (f) Fig. S9 from the previous version, reproduced above as Fig. R4, is replaced with Fig. S10, reproduced above as Fig. R5, with evaluations calculated via new cluster enrichment vectors per individual with updated prediction labels for C_{VAE} clusters.

- (g) Fig. S10 here reproduced as Fig. R10 from the previous version of the manuscript is replaced with Fig. S11, here reproduced as Fig. 9, with evaluations calculated via new cluster enrichment vectors per individual with updated prediction labels for C_{HFE} clusters.
- (h) Additional 2-way comparisons introduced in Fig. S10 from the previous version of the manuscript, reproduced as Fig. R10, were validated and a new supplementary figure is added as Fig. S12 in the revised manuscript and shown above as Fig. R12 with updated predictions for different disease cohorts using C_{HFE} clusters.
-

6. How do you explain the observed changes in nuclear size and mechanical properties of PBMCs in T2DM? Have you controlled for differences in cell subset composition? For example, increased monocytes, dendritic cells, or residual neutrophils could contribute to these observations (besides the cell activation that the authors show).
-

Author reply: We did not prospectively gate PBMC subpopulations for the morphometric analysis, so differences in cell composition (e.g., monocytes, dendritic cells, residual granulocytes) could contribute to the phenotype. Nevertheless, the increase in nuclear area in T2DM that persists across strata of activation (CD25) and heterochromatin load (H3K9Me3) (Fig. S14D–E), and pathology-specific nuclear size shifts that are consistent across donors (Fig. R6B), indicate a cohort-wide shift rather than an isolated subset effect. We suggest that these changes in nuclear shape and chromatin organization are associated with activation and metabolic stress in T2DM [30, 4]. Loss of local H3K9me3 repression at NF- κ B and SASP genes can lead to gene-specific chromatin decondensation and altered mechanical properties [48]. At the same time, global accumulation of heterochromatin (SAHF-like foci) increases condensation in specific compartments [53].

7. In Fig. 5 and S12, the authors conclude: “Together, these observations provide a functional basis for the use of chromatin architecture as a biomarker to track T2DM-related immune alterations.” However, the correlation of nuclear areas with Lamin A/C, H3K9Me3, and CD25 expression does not in itself establish a functional role. Please clarify this point and avoid overinterpretation.
-

Author reply: We agree with the reviewer’s comment and have revised the statement to ensure it aligns strictly with what can be supported by our data. The updated text avoids implying a direct functional role and instead emphasizes that the observed decrease in Lamin expression, chromatin condensation, and activation in individuals with diabetes is consistent with the chromatin state differences identified in PBMC subpopulations through live chromatin imaging.

Revisions: **We have modified the statement that appears on line 178 in the previous version of the manuscript:**

Together, these observations provide a functional basis for the use of chromatin architecture as a biomarker to track T2DM-related immune alterations.

We have replaced this with the following statement on line 253 in the revised

version of the manuscript:

These results reveal alterations in nuclear mechanics and chromatin condensation states of PBMCs in individuals with T2DM, suggesting that chromatin architecture could serve as a sensitive indicator of diabetes-associated immune dysfunction.

8. The authors describe the evidence as “alterations during T2DM progression” throughout the manuscript. Since this is a cross-sectional study, no conclusions about disease progression can be drawn. Please revise the language to avoid implying longitudinal data.
-

Author reply: We have rephrased the wording across the entire manuscript to reflect that the presented study is an cross-sectional and not longitudinal.

Revisions: We have made the following changes to reflect the above details:

Previous version at Line 44: In this study, we selected a cohort of 57 individuals spanning healthy, prediabetic, and diabetic stages.

Revised version at Line 64: In this cross-sectional study, we selected a cohort of 57 individuals spanning healthy, prediabetic, and diabetic stages.

Previous version at Line 46: Our findings indicate that specific PBMC subpopulations reveal patient-specific alterations in immune cells as T2DM advances.

Revised version at Line 70: Our findings indicate that specific PBMC subpopulations reveal patient-specific alterations in immune cells in individuals with T2DM compared to prediabetes and healthy controls.

Previous version at Line 79: Using a microfluidic chromatin imaging assay, we obtained 3D confocal images of Hoechst-stained live PBMCs nuclei from healthy, prediabetic, and diabetic individuals, which exhibited variability in nuclear count.

Revised version at Line 103: Using a microfluidic chromatin imaging assay, we obtained 3D confocal images of Hoechst-stained live PBMCs nuclei from individuals at various stages of T2DM progression, which exhibited variability in nuclear count

Previous version at line 97: Exhaustive VAE embedding of nuclear images from individual patients and at different stages of T2DM progression after sub-sampling is presented in Fig. S6D, S7A, S7B, S8A, and S8B.

Revised version at line 121: Exhaustive VAE embedding of nuclear images from individual patients and from healthy, prediabetes, and diabetes after sub-sampling is presented in Fig. S6D, S7A, S7B, S8A, and S8B.

Previous version at line 113: Taken together these results suggest that the subcluster enrichment of feature-based embeddings of chromatin images can be used to differentiate the stages of disease progression.

Revised version at line 158: Taken together, these results suggest that the subcluster enrichment of feature-based embeddings of chromatin images can be used to differentiate between healthy, prediabetic and diabetic cohort.

Previous version at line 123: We evaluated individual patients’ enrichment profiles across

various stages of disease progression, as shown in Fig. S9A and S9B.

Revised version at line 204: We evaluated individual patients' enrichment profiles across healthy, prediabetes, and diabetes, as shown in Fig. S9A and S9B

Previous version at line 137: These findings suggest that the quantitative representation of chromatin organization in PBMCs, using descriptive features, can provide clinically relevant predictions of immune-related alterations associated with T2DM progression.

Revised version at line 213: These findings suggest that the quantitative representation of chromatin organization in PBMCs, using descriptive features, can provide clinically relevant predictions of immune-related alterations in individuals with T2DM.

-
9. The authors conclude that “specific immune cell subpopulations (Clusters 1 and 6) are enriched in patients with T2DM and healthy individuals.” Could the authors describe these populations phenotypically (e.g., by flow cytometry markers or transcriptomic signatures)? Knowing which clusters are enriched is less informative without understanding their phenotypic and functional properties.
-

Author reply: We would like to clarify that the primary aim of the present study is to show that chromatin imaging of peripheral blood cells is sensitive and specific to T2D-associated alterations, and that these image-derived features can *complement* established immune dysregulation biomarkers (e.g., IL-6, CRP). Because our cohort does not include matched immunophenotyping or transcriptomic measurements, we cannot ascribe the nine chromatin clusters to canonical PBMC lineages (e.g., T cells, monocytes) with confidence. We view these experiments as a natural next step but beyond the scope of the current manuscript. We have clarified this limitation in the revised text.

Revisions:

We have added this text on line 331 of the revised manuscript:

Moreover, our approach does not provide detailed immune cell subpopulation resolution, nor did we perform functional assays (e.g., cytokine secretion, T cell or macrophage activation studies) to mechanistically link chromatin architecture to immune dysfunction. Future studies integrating chromatin imaging with single-cell sequencing or proteomics will be critical to validate and extend these findings.

Discussion:

10. The authors state that the proposed live chromatin imaging provides a clinically relevant readout of immune alterations. However, no detailed cell subset characterization is possible with this technology. What is the clinical relevance of detecting these cell clusters without phenotypic information, especially in the context of POC technologies? Furthermore, the authors mention that inflammatory biomarkers show significant heterogeneity across patients; however, no data on the heterogeneity of the assay results are provided. Also, the cohort size is relatively small; please discuss this limitation.
-

Author reply: We thank the reviewer for these insightful comments. We agree that live

chromatin imaging does not provide specific characterization of gene/protein expression comparable to flow cytometry or single-cell multi-omics. Our intent is not to replace these high-dimensional approaches but to demonstrate that chromatin architecture can serve as a rapid, label-free, and functional readout of immune alterations with potential point-of-care applicability. Importantly, chromatin organization reflects integrated transcriptional and metabolic states, thereby providing clinically relevant information even without explicit phenotypic annotation. We also acknowledge the heterogeneity of immune alterations across patients. While our proof-of-concept study focused on identifying consistent group-level differences, the directionality of changes (e.g., decreased Lamin A/C, increased chromatin condensation in T2DM) was reproducible across individuals. We agree that larger studies will be required to quantify patient-to-patient variability and assess correlations with clinical parameters such as HbA1c and BMI. Finally, we recognize that the cohort size is relatively modest; however, the fact that statistically significant differences were detected underscores the robustness of the assay. We view this work as a feasibility study that lays the foundation for future validation in larger, independent cohorts.

Revisions:

We have added this text on line 326 of the revised manuscript:

Nevertheless, several limitations should be noted. For instance, our algorithms and experiments were developed and tested in a single cohort, and independent validation in larger, multi-center cohorts will be required to establish reproducibility and generalizability.

11. The authors mention the cost-effectiveness of their technology. It would be useful to compare costs with established biomarker assays to better contextualize this claim.
-

Author reply: We acknowledge that the comparative cost analysis is useful in comparing our approach with established biomarkers. Clinically relevant assays to measure HbA1c and CRP, low-cost assays [15, 22], while IL-6 is substantially more expensive [8]. Our approach remains inexpensive and uses low-cost reagents such as Hoechst dye and microfluidics devices [21], suggesting material costs could be comparable with scalability. The current limitation requiring research lab infrastructure (e.g. centrifuge, confocal microscope, workflow complexity) can be addressed with the comparative automated screening workflows.

Revisions: **We have added the following sentences to the discussion section on line 317 of the revised manuscript:**

From a cost perspective, HbA1c and CRP are low-cost assays [15, 22], while IL-6 is substantially more expensive [8] (Table S3). Our approach remains inexpensive and uses low cost reagents such as Hoechst dye and microfluidics devices [21], suggesting material costs could be comparable with scalability (Table S3).

We have included a section in the following table in the Supplementary section of the Revised manuscript:

Dimension	HbA1c	CRP / IL-6	Chromatin imaging (this work)
Cost per test (relative)	Low (\$8 – \$16) [15]	CRP: Low (\$12 – \$16)[22]; IL-6: Moderate: (\$100 – \$300) [8]	Low (Estimated with Scalability): ≤ \$10; includes low cost reagents like Hoechst (\$143/10 mL Thermo Fischer) & Microfluidics Devices [21]

Table 4: Comparison of HbA1c, CRP/IL-6, and chromatin imaging assay across diagnostic and practical dimensions.

References

- [1] M Abramiuk, I Dymanowska-Dyjak, E Grywalska, and G Polak. P-352 evaluation of cd25 and cd69 activation markers expression on peripheral blood cells subpopulations in endometriosis patients. *Human Reproduction*, 38(Supplement_1):dead093–710, 2023.
- [2] Ramzi A Ajjan, Tadej Battelino, Xavier Cos, Stefano Del Prato, Jean-Christophe Philips, Laurent Meyer, Jochen Seufert, and Samuel Seidu. Continuous glucose monitoring for the routine care of type 2 diabetes mellitus. *Nature Reviews Endocrinology*, 20(7):426–440, 2024.
- [3] CM Bennett, M Guo, and SC Dharmage. Hba1c as a screening tool for detection of type 2 diabetes: a systematic review. *Diabetic medicine*, 24(4):333–343, 2007.
- [4] B Cai, J Zhang, M Zhang, L Li, W Feng, Z An, and L Wang. Micro-inflammation characterized by disturbed treg/teff balance with increasing sil-2r in patients with type 2 diabetes. *Experimental and Clinical Endocrinology & Diabetes*, 121(04):214–219, 2013.
- [5] Hui Chen, Xiangrong Ren, Nanying Liao, and Feng Wen. Th17 cell frequency and il-17a concentrations in peripheral blood mononuclear cells and vitreous fluid from patients with diabetic retinopathy. *Journal of International Medical Research*, 44(6):1403–1413, 2016.
- [6] Gholamreza Daryabor, Mohamad Reza Atashzar, Dieter Kabelitz, Seppo Meri, and Kurosh Kalantar. The effects of type 2 diabetes mellitus on organ metabolism and the immune system. *Frontiers in immunology*, 11:1582, 2020.
- [7] Marc Y Donath and Steven E Shoelson. Type 2 diabetes as an inflammatory disease. *Nature reviews immunology*, 11(2):98–107, 2011.
- [8] FindLabtest.com, 2025. Accessed: 2025-09-04.
- [9] Guido Freckmann, Sina Buck, Delia Waldenmaier, Bernhard Kulzer, Oliver Schnell, Ulrich Gelchsheimer, Ralph Ziegler, and Lutz Heinemann. Insulin pump therapy for patients with type 2 diabetes mellitus: Evidence, current barriers, and new technologies. *Journal of Diabetes Science and Technology*, 15(4):901–915, 2021. PMID: 32476471.
- [10] Anna Giovenzana, Eugenia Bezzecchi, Anita Bichisecchi, Sara Cardellini, Francesca Raggogna, Federica Pedica, Federica Invernizzi, Luigi Di Filippo, Valentina Tomajer, Francesca Aleotti, et al. Fat-to-blood recirculation of partially dysfunctional pd-1+ cd4 tconv cells is associated with dysglycemia in human obesity. *Iscience*, 27(3), 2024.
- [11] Anna Giovenzana, Valentina Codazzi, Michele Pandolfo, and Alessandra Petrelli. T cell trafficking in human chronic inflammatory diseases. *Iscience*, 27(8), 2024.
- [12] Vera Grossmann, Volker H Schmitt, Tanja Zeller, Marina Panova-Noeva, Andreas Schulz, Dagmar Laubert-Reh, Claus Juenger, Renate B Schnabel, Tobias GJ Abt, Rafael Laskowski, et al. Profile of the immune and inflammatory response in individuals with prediabetes and type 2 diabetes. *Diabetes care*, 38(7):1356–1364, 2015.

- [13] Daeon Gu, Jinyeong Lim, Kyung Yeon Han, In-Ho Seo, Jae Hwan Jee, Soo Jin Cho, Yoon Ho Choi, Sung Chul Choi, Jang Hyun Koh, Jin-Young Lee, et al. Single-cell analysis of human pbmcs in healthy and type 2 diabetes populations: dysregulated immune networks in type 2 diabetes unveiled through single-cell profiling. *Frontiers in Endocrinology*, 15:1397661, 2024.
- [14] Rajshikhar Gupta, Paulina Schärer, Yawen Liao, Bibhas Roy, Roger M Benoit, and GV Shivashankar. Regulation of p65 nuclear localization and chromatin states by compressive force. *Molecular Biology of the Cell*, 36(4):ar37, 2025.
- [15] Shaivya Gupta, U Jain, N Chauhan, et al. Laboratory diagnosis of hba1c: a review. *J Nanomed Res*, 5(4):00120, 2017.
- [16] Robert Haas, Joanne Smith, Vidalba Rocher-Ros, Suchita Nadkarni, Trinidad Montero-Melendez, Fulvio D’acquisto, Elliot J Bland, Michele Bombardieri, Costantino Pitzalis, Mauro Perretti, et al. Lactate regulates metabolic and pro-inflammatory circuits in control of t cell migration and effector functions. *PLoS biology*, 13(7):e1002202, 2015.
- [17] Thomas Hägglöf, Carlo Vanz, Abigail Kumagai, Elizabeth Dudley, Vanessa Ortega, McKenzie Siller, Raksha Parthasarathy, Josh Keegan, Abigail Koenigs, Travis Shute, et al. T-bet+ b cells accumulate in adipose tissue and exacerbate metabolic disorder during obesity. *Cell metabolism*, 34(8):1121–1136, 2022.
- [18] Wei He, Ting Yuan, and Kathrin Maedler. Macrophage-associated pro-inflammatory state in human islets from obese individuals. *Nutrition & diabetes*, 9(1):36, 2019.
- [19] Iren D Hjellestad, Marianne C Astor, Roy M Nilsen, Eirik Sjøfteland, and Torbjørn Jonung. Hba1c versus oral glucose tolerance test as a method to diagnose diabetes mellitus in vascular surgery patients. *Cardiovascular diabetology*, 12(1):79, 2013.
- [20] Gökhan S Hotamisligil. Foundations of immunometabolism and implications for metabolic health and disease. *Immunity*, 47(3):406–420, 2017.
- [21] <https://www.ufluidix.com/pricing/>. ufluidix pricing information, 2025. Accessed: 2025-09-04.
- [22] Rachael Hunter. Cost-effectiveness of point-of-care c-reactive protein tests for respiratory tract infection in primary care in england. *Advances in therapy*, 32(1):69–85, 2015.
- [23] Juan Jin, Longqiang Wang, Yongjun Liu, Wenfang He, Danna Zheng, Yinhua Ni, and Qiang He. Depiction of immune heterogeneity of peripheral blood from patients with type ii diabetic nephropathy based on mass cytometry. *Frontiers in Endocrinology*, 13:1018608, 2023.
- [24] EYM Lau, EC Carroll, LA Callender, GA Hood, V Berryman, M Patrick, S Finer, GA Hitman, GL Ackland, and SM Henson. Type 2 diabetes is associated with the accumulation of senescent t cells. *Clinical & Experimental Immunology*, 197(2):205–213, 2019.
- [25] L Lei, L Cui, Y Mao, X Zhang, Q Jiang, S Dong, and Y Wang. Augmented cd25 and cd69 expression on circulating cd8+ t cells in type 2 diabetes mellitus with albuminuria. *Diabetes & metabolism*, 43(4):382–384, 2017.
- [26] Ge Li, Lanwen Han, Yonghui Wang, Yanglu Zhao, Yu Li, Junling Fu, Ming Li, Shan Gao, and Steven M Willi. Evaluation of ada hba1c criteria in the diagnosis of pre-diabetes and diabetes in a population of chinese adolescents and young adults at high risk for diabetes: a cross-sectional study. *BMJ open*, 8(8):e020665, 2018.

- [27] Chiara Macchi, Annalisa Moregola, Maria Francesca Greco, Monika Svecla, Fabrizia Bonacina, Suveera Dhup, Rajesh Kumar Dadhich, Matteo Audano, Pierre Sonveaux, Claudio Mauro, et al. Monocarboxylate transporter 1 deficiency impacts cd8+ t lymphocytes proliferation and recruitment to adipose tissue during obesity. *IScience*, 25(6), 2022.
- [28] Igor Mandric, Tommer Schwarz, Arunabha Majumdar, Kangcheng Hou, Leah Briscoe, Richard Perez, Meena Subramaniam, Christoph Hafemeister, Rahul Satija, Chun Jimmie Ye, et al. Optimized design of single-cell rna sequencing experiments for cell-type-specific eqtl analysis. *Nature communications*, 11(1):5504, 2020.
- [29] James B Meigs. Multiple biomarker prediction of type 2 diabetes, 2009.
- [30] Feng Miao, Xiwei Wu, Lingxiao Zhang, Yate-Ching Yuan, Arthur D Riggs, and Rama Natarajan. Genome-wide analysis of histone lysine methylation variations caused by diabetic conditions in human monocytes. *Journal of Biological Chemistry*, 282(18):13854–13863, 2007.
- [31] Stefano Monti, Pablo Tamayo, Jill Mesirov, and Todd Golub. Consensus clustering: a resampling-based method for class discovery and visualization of gene expression microarray data. *Machine learning*, 52(1):91–118, 2003.
- [32] HW Nam, YJ Cho, JA Lim, SJ Kim, H Kim, SY Sim, and DG Lim. Functional status of immune cells in patients with long-lasting type 2 diabetes mellitus. *Clinical & Experimental Immunology*, 194(1):125–136, 2018.
- [33] Ha Thi Nga, Thi Linh Nguyen, and Hyon-Seung Yi. T-cell senescence in human metabolic diseases. *Diabetes & Metabolism Journal*, 48(5):864–881, 2024.
- [34] Ichiro Nojima, Shingo Eikawa, Nahoko Tomonobu, Yoshiko Hada, Nobuo Kajitani, Sanae Teshigawara, Satoshi Miyamoto, Atsuhito Tone, Haruhito A Uchida, Atsuko Nakatsuka, et al. Dysfunction of cd8+ pd-1+ t cells in type 2 diabetes caused by the impairment of metabolism-immune axis. *Scientific reports*, 10(1):14928, 2020.
- [35] Olivia Osborn and Jerrold M Olefsky. The cellular and signaling networks linking the immune system and metabolism in disease. *Nature medicine*, 18(3):363–374, 2012.
- [36] Chanchira Phosat, Pornpimol Panprathip, Noppanath Chumpathat, Pattaneeya Prangthip, Narisara Chantratita, Ngamphol Soonthornworasiri, Somchai Puduang, and Karunee Kwanbunjan. Elevated c-reactive protein, interleukin 6, tumor necrosis factor alpha and glycemic load associated with type 2 diabetes mellitus in rural thais: a cross-sectional study. *BMC endocrine disorders*, 17:1–8, 2017.
- [37] Aruna D Pradhan, JoAnn E Manson, Nader Rifai, Julie E Buring, and Paul M Ridker. C-reactive protein, interleukin 6, and risk of developing type 2 diabetes mellitus. *jama*, 286(3):327–334, 2001.
- [38] Sukanya Raghuraman, Ida Donkin, Soetkin Versteyhe, Romain Barrès, and David Simar. The emerging role of epigenetics in inflammation and immunometabolism. *Trends in Endocrinology and Metabolism*, 27(11):782–795, 2016.
- [39] IM Rea, SE McNerlan, and HD Alexander. Cd69, cd25, and hla-dr activation antigen expression on cd3+ lymphocytes and relationship to serum tnf- α , ifn- γ , and sil-2r levels in aging. *Experimental gerontology*, 34(1):79–93, 1999.

- [40] Marpadga A Reddy, Vishnu Amaram, Sadhan Das, Vinay Singh Tanwar, Rituparna Ganguly, Mei Wang, Linda Lanting, Lingxiao Zhang, Maryam Abdollahi, Zhuo Chen, et al. Incrna drair is downregulated in diabetic monocytes and modulates the inflammatory phenotype via epigenetic mechanisms. *JCI insight*, 6(11):e143289, 2021.
- [41] Blanca I Restrepo, Marcel Twahirwa, and Chinnaswamy Jagannath. Hyperglycemia and dyslipidemia: Reduced hla-dr expression in monocyte subpopulations from diabetes patients. *Human immunology*, 82(2):124–129, 2021.
- [42] Sara SantaCruz-Calvo, Leena Bharath, Gabriella Pugh, Lucia SantaCruz-Calvo, Raji Ramesh Lenin, Jenny Lutshumba, Rui Liu, Adam D Bachstetter, Beibei Zhu, and Barbara S Nikolajczyk. Adaptive immune cells shape obesity-associated type 2 diabetes mellitus and less prominent comorbidities. *Nature Reviews Endocrinology*, 18(1):23–42, 2022.
- [43] Byeong Chang Sim, Yea Eun Kang, Sun Kyoung You, Seong Eun Lee, Ha Thi Nga, Ho Yeop Lee, Thi Linh Nguyen, Ji Sun Moon, Jingwen Tian, Hyo Ju Jang, et al. Hepatic t-cell senescence and exhaustion are implicated in the progression of fatty liver disease in patients with type 2 diabetes and mouse model with nonalcoholic steatohepatitis. *Cell Death & Disease*, 14(9):618, 2023.
- [44] Amanda Smythers and Benjamin Orsburn. The current economics and throughput of single cell proteomics by liquid chromatography mass spectrometry. 2024.
- [45] Monica Alejandra Valtierra-Alvarado, JE Castañeda Delgado, Sandra Iveth Ramírez-Talavera, Geanncarlo Lugo-Villarino, Fátima Dueñas-Arteaga, Anahí Lugo-Sánchez, MS Adame-Villalpando, Bruno Rivas-Santiago, J Enciso-Moreno, and Carmen Judith Serano. Type 2 diabetes mellitus metabolic control correlates with the phenotype of human monocytes and monocyte-derived macrophages. *Journal of Diabetes and its Complications*, 34(11):107708, 2020.
- [46] Saradha Venkatachalapathy, Doorgesh Sharma Jokhun, and GV Shivashankar. Multivariate analysis reveals activation-primed fibroblast geometric states in engineered 3d tumor microenvironments. *Molecular Biology of the Cell*, 31(8):803–812, 2020.
- [47] Louisa M Villeneuve, Marpadga A Reddy, Linda L Lanting, Mei Wang, Li Meng, and Rama Natarajan. Epigenetic histone h3 lysine 9 methylation in metabolic memory and inflammatory phenotype of vascular smooth muscle cells in diabetes. *Proceedings of the National Academy of Sciences*, 105(26):9047–9052, 2008.
- [48] Maria Cristina Vinci, Sarah Costantino, Giulia Damiano, Erica Rurali, Raffaella Rinaldi, Vera Vigorelli, Annalisa Sforza, Ermes Carulli, Sergio Pirola, Giorgio Mastroiacovo, et al. Persistent epigenetic signals propel a senescence-associated secretory phenotype and trained innate immunity in cd34+ hematopoietic stem cells from diabetic patients. *Cardiovascular Diabetology*, 23(1):107, 2024.
- [49] Hua Wang, Wajahat Mehal, Laura E Nagy, and Yaron Rotman. Immunological mechanisms and therapeutic targets of fatty liver diseases. *Cellular & Molecular Immunology*, 18(1):73–91, 2021.
- [50] Yan Wang, Shu-yan Zhao, Yong-chun Wang, Jia Xu, and Jie Wang. The immune-inflammation factor is associated with diabetic nephropathy: evidence from nhanes 2013–2018 and geo database. *Scientific Reports*, 14(1):17760, 2024.

- [51] Ying Wang, Mengwei Li, Lin Chen, Huan Bian, Xiangying Chen, Huilin Zheng, Peiwei Yang, Quan Chen, and Hanmei Xu. Natural killer cell-derived exosomal mir-1249-3p attenuates insulin resistance and inflammation in mouse models of type 2 diabetes. *Signal transduction and targeted therapy*, 6(1):409, 2021.
- [52] Leonel Witcoski Junior, Jordana Dinorá de Lima, Amanda Girardi Somensi, Lucas Brito de Souza Santos, Giulia Leonel Paschoal, Thalita Suemy Uada, Thais Sibioni Berti Bastos, André Guilherme Portela de Paula, Rebeca Bosso Dos Santos Luz, Andressa Pacheco Czaikovski, et al. Metabolic reprogramming of macrophages in the context of type 2 diabetes. *European Journal of Medical Research*, 29(1):497, 2024.
- [53] Yan-Lin Wu, Zheng-Jun Lin, Chang-Chun Li, Xiao Lin, Su-Kang Shan, Bei Guo, Ming-Hui Zheng, Fuxingzi Li, Ling-Qing Yuan, and Zhi-hong Li. Epigenetic regulation in metabolic diseases: mechanisms and advances in clinical study. *Signal Transduction and Targeted Therapy*, 8(1):98, 2023.
- [54] Hyon-Seung Yi, So Yeon Kim, Jung Tae Kim, Young-Sun Lee, Ji Sun Moon, Mingyo Kim, Yea Eun Kang, Kyong Hye Joung, Ju Hee Lee, Hyun Jin Kim, et al. T-cell senescence contributes to abnormal glucose homeostasis in humans and mice. *Cell death & disease*, 10(3):249, 2019.
- [55] Xinyi Zhang, Saradha Venkatachalapathy, Daniel Paysan, Paulina Schaerer, Claudio Tripodo, Caroline Uhler, and GV Shivashankar. Unsupervised representation learning of chromatin images identifies changes in cell state and tissue organization in dcis. *Nature Communications*, 15(1):6112, 2024.
- [56] Yixuan Zhao and Rensong Yue. White adipose tissue in type 2 diabetes and the effect of antidiabetic drugs. *Diabetology & Metabolic Syndrome*, 17(1):116, 2025.